# Antigen presentation by B cells enables epitope spreading across an MHC barrier

Cecilia Fahlquist-Hagert [1,14], Thomas R. Wittenborn [1,14], Ewa Terczyńska-Dyla[1], Kristian Savstrup Kastberg [1], Emily Yang [2], Alysa Nicole Rallistan[2], Quinton Raymond Markett [2], Gudrun Winther [1], Sofie Fonager[1], Lasse F. Voss [1,3], Mathias K. Pedersen [1], Nina van Campen [1,4], Alexey Ferapontov[1,5], Lisbeth Jensen[1], Jinrong Huang [6,7], John D. Nieland [8], Cees E. van der Poel[9,13], Johan Palmfeldt [10], Michael C. Carroll [9], Paul J. Utz [2,11], Yonglun Luo [6,7,12], Lin Lin [6,12] & Søren E. Degn [1,5] ✉

Circumstantial evidence suggests that B cells may instruct T cells to break tolerance. Here, to test this hypothesis, we used a murine model in which a single B cell clone precipitates an autoreactive response resembling systemic lupus erythematosus (SLE). The initiating clone did not need to enter germinal centers to precipitate epitope spreading. Rather, it localized to extrafollicular splenic bridging channels early in the response. Autoantibody produced by the initiating clone was not sufficient to drive the autoreactive response. Subsequent epitope spreading depended on antigen presentation and was compartmentalized by major histocompatibility complex (MHC). B cells carrying two MHC haplotypes could bridge the MHC barrier between B cells that did not share MHC. Thus, B cells directly relay autoreactivity between two separate compartments of MHC-restricted T cells, leading to inclusion of distinct B cell populations in germinal centers. Our findings demonstrate that B cells initiate and propagate the autoimmune response.

Autoimmune diseases are a complex and heterogenous group of disorders representing a growing societal burden[1]. Although the understanding of many autoimmune diseases has expanded significantly in recent years, in most cases the etiology remains obscure. However, mounting evidence suggests that antigen presentation by B cells can play a fundamental role in the initiation of numerous autoimmune diseases, such as Rheumatoid Arthritis[2], Type I Diabetes Mellitus[3,4], and

Multiple Sclerosis[5]. Whereas macrophages and dendritic cells internalize antigen through phagocytosis or pinocytosis, B cells capture antigen with high specificity through their B-cell receptors. Consequently, they are potent antigen-presenting cells, capable of focusing the antigenic response, even when antigen is present at low concentrations[6–8]. Antigen-specific B cells internalize antigen 10,000-fold more efficiently than non-specific cells, highlighting this

[1]Laboratory for Lymphocyte Biology, Department of Biomedicine, Aarhus University, Aarhus C, Denmark. [2]Department of Medicine, Division of Immunology and Rheumatology, Stanford University, Stanford, CA, USA. [3]Department of Health Technology, Technical University of Denmark, Lyngby, Denmark. [4]Department of Biomedical Sciences, Radboud University Medical Center, Nijmegen, The Netherlands. [5]CellPAT Center for Cellular Signal Patterns, iNANO, Aarhus University, Aarhus C, Denmark. [6]DREAM Laboratory for Applied Genome Technologies, Department of Biomedicine, Aarhus University, Aarhus C, Denmark. [7]Lars Bolund Institute of Regenerative Medicine, Qingdao-Europe Advanced Institute for Life Sciences, BGI-Qingdao, BGI-Shenzhen, Shenzhen, China. [8]Department of Health Science and Technology, Faculty of Medicine, Aalborg University, Gistrup, Denmark. [9]Program in Cellular and Molecular Medicine, Boston Children's Hospital, Boston, MA, USA. [10]Research Unit for Molecular Medicine, Department of Clinical Medicine, Aarhus University, Aarhus N, Denmark. [11]Institute for Immunity, Transplantation and Infection, Stanford University School of Medicine, Stanford, CA, USA. [12]Steno Diabetes Center Aarhus, Aarhus University Hospital, Aarhus N, Denmark. [13]Present address: Dragonfly Therapeutics, Waltham, MA, USA. [14]These authors contributed equally: Cecilia Fahlquist-Hagert, Thomas R. Wittenborn. ✉e-mail: sdegn@biomed.au.dk

capacity[9,10]. This may be particularly important in autoimmune diseases, where autoantigen initially may be available at low amounts[11].

Systemic lupus erythematosus (SLE), is a systemic autoimmune disease, characterized by the presence of affinity-matured, class-switched autoantibodies[12]. It has been a long-standing observation that the breadth of targeted autoantigens increases over time, potentially starting decades before clinical presentation[13]. This process is termed epitope spreading, and it can be defined as the diversification of epitope specificity from an initial focused, dominant epitope-specific immune response[14]. While epitope spreading may contribute to robust protection against pathogens, it is also thought to play a central role in the pathogenic process of autoimmune diseases by expanding the range of cellular components and tissues targeted by the immune system[15,16]. However, it remains unclear how tolerance is broken and how epitope spreading is initiated in SLE.

Highly affine antibodies are generated by B cells in micro-anatomical structures termed germinal centers (GCs)[17]. Here, B-cell clones cycle through consecutive rounds of division and somatic hypermutation (SHM) in the dark zone, and selection based on antigen affinity in the light zone[18]. B cells acquire antigens from follicular dendritic cells through their B-cell receptors (BCRs) and present derived peptides on major histocompatibility complex class II (MHCII) to solicit help from cognate T follicular helper cells[19]. T-cell help in turn determines the selection of B cells for return to the dark zone[20–22]. B cells that do not receive help undergo apoptosis, whereas B cells that do receive help return to the dark zone, where they may undergo another round of division and SHM. Thus, iterative cycles of diversity generation and selection shape the B-cell and antibody repertoire over time, from initiation of the GC response until clearance of the eliciting antigen[23].

The provision of T-cell help to B cells via cognate TCR:peptide-MHC (pMHC) interactions necessitates that antigenic complexes taken up by BCR-mediated endocytosis contain peptides which are recognized as foreign in the context of MHCII. This requirement, termed linked recognition, is thought to be critical to restrict the de novo emergence of autoreactive specificities in the GC. Yet in autoimmunity, it has been found that somatically mutated and affinity-matured antibodies to self-antigens are prevalent[24]. Although such antibodies can be generated outside GCs to a lower extent[25], many autoimmune diseases, such as SLE, display spontaneous GC responses. It is generally believed that the GC process is co-opted to generate affinity matured autoantibodies targeting the host's own antigens[26], but much less is known about the nature of autoreactive GC responses than those directed towards foreign model antigens.

We previously developed a mixed chimera model of autoreactivity displaying spontaneous GCs and epitope spreading reminiscent of that observed in SLE[27]. A B-cell clone carrying a knock-in of an autoreactive BCR (clone 564Igi), initiates an autoimmune process, which spreads to the repertoire derived from one or more wild-type (WT) compartments through inclusion of proto-autoreactive B cells from the naive repertoire into GC responses[27,28]. Two, not necessarily mutually exclusive, mechanisms can be invoked to explain the ability of the 564Igi clone to break tolerance and initiate epitope spreading to the WT compartment. One possibility is that the 564Igi clone establishes an environment that is permissive for autoreactivity. For example, the 564Igi clone produces autoantibodies, which form immune complexes that cause inflammation and get taken up by myeloid cells, which in turn break T-cell tolerance. The other is that the 564Igi clone acts as an antigen-presenting cell, directly driving a break-of-tolerance in the T-cell compartment. An important functional distinction between the two mechanisms is that the latter would limit the epitope spreading to T cells restricted for the MHC haplotype of the 564Igi cells, whereas the former would allow epitope spreading globally.

Here, we leveraged the mixed chimera model to explore the basic mechanisms governing the initial break-of-tolerance and subsequent epitope spreading, in a setting where competing WT compartments have different MHC haplotypes, enabling or restricting their communication with the autoreactive driver compartment. We show that the initial break-of-tolerance occurs outside GCs and demonstrate that a cellular relay mechanism subsequently allows MHC-dependent autoreactive epitope spreading within GCs.

## Results

### The initiating autoreactive B-cell clone does not need to enter GCs to break tolerance

We previously reported that a single autoreactive B-cell clone, clone 564Igi, could break tolerance and initiate epitope spreading[27]. A likely explanation was that this clone could seed autoreactive GCs. However, in our mixed chimera model, the spontaneous GCs observed 6 weeks post reconstitution were almost exclusively composed of WT-derived clones[27]. This was surprising, because clone 564Igi harbors an autoantibody knock-in that was affinity matured[29] and hence should carry a selective advantage over WT clones in affinity-driven competition in GCs[17]. Accordingly, we speculated that at an earlier time point, GCs could be dominated by the initiating clone, and that it was only subsequently outcompeted by more diverse clones derived from the WT repertoire.

To address this, we undertook a kinetic analysis of GC formation in 564Igi H$^{ki/ki}$K$^{ki/ki}$ (homozygous for the heavy and light chain knock-in) bone marrow (BM) chimeras or 564Igi H$^{ki/ki}$K$^{wt/wt}$ (heavy chain homozygous knock-in only, kappa chain wild-type) control chimeras, analyzing at weeks 2, 3, 4 and 6 (Fig. 1A). We observed a surge of B cells at 2 weeks following reconstitution, dropping to normal levels (gray area) at 3-4 weeks post reconstitution (Fig. 1B). There was an even more marked surge of Idiotype positive cells (Id+, carrying the B-cell receptor defined by the heavy and light chain knock-ins) in 564Igi

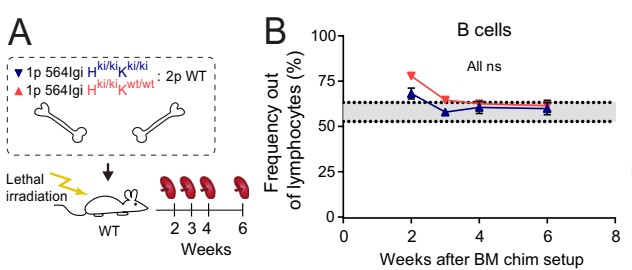
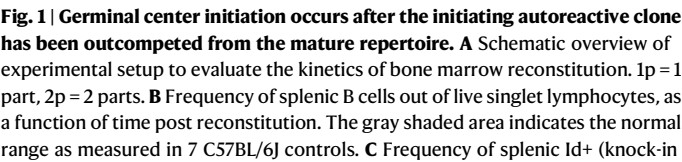
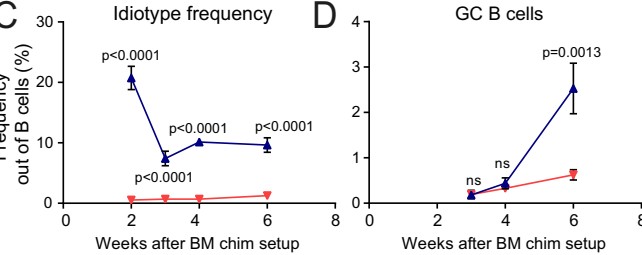

**Fig. 1 | Germinal center initiation occurs after the initiating autoreactive clone has been outcompeted from the mature repertoire.** **A** Schematic overview of experimental setup to evaluate the kinetics of bone marrow reconstitution. 1p = 1 part, 2p = 2 parts. **B** Frequency of splenic B cells out of live singlet lymphocytes, as a function of time post reconstitution. The gray shaded area indicates the normal range as measured in 7 C57BL/6J controls. **C** Frequency of splenic Id+ (knock-in receptor) B cells out of total B cells, as a function of time post reconstitution. **D** Frequency of splenic GC B cells out of total B cells, as a function of time post reconstitution. **B–D** show Mean ± SEM for $n = 3$, 1 (week 2), 7, 7 (week 3), 3, 4 (week 4), 7, 5 (week 6) 564Igi H$^{ki/ki}$K$^{ki/ki}$ and 564Igi H$^{ki/ki}$K$^{wt/wt}$ chimeras, respectively. Statistical significance shown for two-way ANOVA with Šidák's post-test. ns = $p > 0.05$.

mixed chimeras, which also readjusted at 3-4 weeks (Fig. 1C). In control chimeras, there was a constant low background level of Id+ cells. The splenic GC frequency remained low at 4 weeks, then increased by 6 weeks in 564Igi mixed chimeras, but not in controls (Fig. 1D). Taken together, our data suggested that GCs began to arise around 3 weeks, but a full-blown GC phenotype did not manifest until sometime between 4-6 weeks. The drop-off in Id+ B cells prior to the emergence of GCs suggested that these cells were unlikely to directly seed GCs.

To more definitively determine whether entry of the autoreactive driver clone into GCs was required for initiation of the autoimmune process, we set up mixed chimeras in which these cells were precluded from entering GCs. This was achieved by crossing the 564Igi line to Aicda-Cre Bcl6$^{flx/flx}$ background, causing a deletion of the GC transcription factor Bcl6 in antigen-experienced B cells. Aicda-Cre+ Bcl6$^{flx/flx}$ 564Igi H$^{ki/ki}$K$^{ki/ki}$ or similar Aicda-Cre negative control BM, both carrying CD45.2 congenic marker, was used for reconstitution of lethally irradiated WT recipients carrying the congenic marker CD45.1, at a 1:2 ratio with WT donors also carrying CD45.1 congenic marker (Fig. 2A). Eight weeks post reconstitution, chimeras displayed comparable levels of CD4 T, CD8 T, and B cells (Fig. 2B–D). Idiotype B cells were also comparable between Cre+ and Cre- chimeras, demonstrating that the GC block did not impact the overall level of B cells carrying the knock-in receptor (Fig. 2E). The levels of GC B cells, plasmablasts and plasma cells (Fig. 2F–H) were also comparable between the Cre+ and Cre- chimeras, in support of the conclusion that the autoreactive driver B cells did not need to enter GCs to precipitate the autoimmune response in the WT compartment.

As a control for the efficacy of the genetic GC block, we also analyzed GC blocked 564Igi H$^{ki/ki}$ K$^{ki/ki}$ Aicda-Cre+ Bcl6$^{flx/flx}$ mice and GC sufficient 564Igi H$^{ki/ki}$ K$^{ki/ki}$ Aicda-Cre- Bcl6$^{flx/flx}$ littermate controls. We compared the GC B-cell levels, idiotype, plasmablast and plasma cell frequencies to those of the bone marrow chimera groups and naive C57BL/6J mice (Supplementary Fig. 1A-E). Naive C57BL/6J mice had noticeable GC B-cell levels only in the MesLN, Cre- 564Igi mice had robust GC responses in all lymphoid tissues, whereas Cre+ 564Igi mice harbored virtually no GC B cells (Supplementary Fig. 1B). As expected, both 564Igi groups had high Id+ frequencies, whereas C57BL/6J had virtually none (Supplementary Fig. 1C). There were no significant differences in plasmablast and plasma cell frequencies between the groups (Supplementary Fig. 1D, E).

Focusing again on the chimeras, the frequency of CD45.2 cells in the B-cell compartment was similar between Cre+ and Cre- groups (Fig. 2I). However, whereas CD45.2 cells were present among GC B cells of Cre- chimeras, they were virtually excluded from GCs of Cre+ chimeras (Fig. 2J), confirming the fidelity of the GC block. Nonetheless, CD45.2 cells contributed similarly to plasmablast and plasma cell pools of Cre+ and Cre- chimeras (Fig. 2K, L), and total levels of anti-dsDNA antibodies were comparable (Fig. 2M). We additionally measured isotype-specific anti-dsDNA antibodies that derived either from the 564Igi compartment (IgG2a) or the wild-type compartment (IgG2c). Again, these were comparable between the groups, although of note the IgG2c levels were very low (Fig. 2M). Furthermore, we measured the level of antibodies carrying the idiotype of the knock-in or possibly related members of the same idiotypic family, and again this indicated comparable levels between the two groups (Fig. S1F–H). Immunofluorescence microscopy of spleen sections confirmed comparable levels of splenic GCs between the two groups (Fig. 2N) and verified the absence of CD45.2 B cells from GCs of Cre+ chimeras (Fig. 2O-S).

Strikingly, by imaging, we observed prominent idiotype (9D11) bright cells and CD138+ cells outside white pulp areas proximal to sites where the marginal zone macrophage (CD169) border was disrupted in both Cre- and Cre+ chimeras (Fig. 2T–W). These anatomical locations corresponded well with previously described extrafollicular splenic bridging channels (EFSBCs)[30].

## Autoantibody derived from the initiating B-cell clone is insufficient to break tolerance

The notion that the initiating autoreactive clone did not need to enter GCs to break tolerance, combined with the robust contribution of autoreactive driver cells to the plasma cell compartment even when blocked from GC entry, raised the possibility that 564Igi B cell differentiation to plasma cells and production of autoantibody could establish an autoreactive environment.

To test this hypothesis, mice were exposed to the 564Igi autoantibody over a time frame comparable to that of the reconstitution period of 564Igi mixed chimeras (Fig. 3A). To ensure a more permissive environment for break-of-tolerance, we used FoxP3-Cre Bcl6$^{flx/flx}$ mice, devoid of T follicular regulatory (Tfr) cells and incapable of upregulating FoxP3 in Tfh cells to shut down GCs[31–34]. Despite this, mice treated with the Idiotypic antibody (564Igi, clone C11) displayed levels of B, CD8 T, CD4 T, GCB, plasmablasts and plasma cells (Fig. 3B-F), which were indistinguishable from those of non-injected controls (no Ab) and mice treated similarly with normal murine IgG, or anti-idiotype antibody (clone 9D11).

We confirmed that FoxP3-Cre Bcl6$^{flx/flx}$ background supported the autoreactive phenotype of 564Igi mice by crossing these lines together to generate Tfr blocked 564Igi H$^{ki/wt}$ K$^{ki/wt}$ FoxP3-Cre Bcl6$^{flx/flx}$ mice, and additionally compared to regular 564Igi H$^{ki/wt}$ K$^{ki/wt}$ mice (Supplementary Fig. 1I–M). Indeed, both these groups displayed robust GC and Id+ B cell frequencies across all lymphoid tissues (Supplementary Fig. 1K, M). Plasmablast/cell levels were high and appeared additionally elevated in the Tfr blocked group (Supplementary Fig. 1L) in agreement with the observation that Tfr regulate plasma cell output from GCs[35].

We also considered the possibility that administration of the autoreactive idiotype could induce an anti-idiotypic response, or conversely, that administration of the anti-idiotype could elicit an idiotypic and thereby potentially autoreactive response. To evaluate the presence of such cells or their secreted products, we employed idiotypic and anti-idiotypic staining of cells derived from the chimeras and solid-phase assay screening of serum antibodies (Fig. 3G). We did not observe any significant levels of idiotypic or anti-idiotypic cells in any of the injected groups (Fig. 3H, I). Sera from the group that received injections of idiotypic antibody harbored a robust level of this clone (Fig. 3J), and conversely for anti-idiotypic antibody in the group that received that clone (Fig. 3K), but no other differences were observed.

Although this strongly suggested that the presence of the autoantibody produced by the initiating clone, even in a permissive environment, was insufficient to precipitate a break-of-tolerance, the possibility remained that the environment did not adequately recapitulate salient features of the mixed chimera setting. To further approximate this, we repeated the experiment in WT BM chimeras (Fig. 3L). The environment should be comparable to that of 564Igi mixed chimeras, with a high cell turnover and transient depletion of mature cell compartments, and the presence of autoantibody capable of forming immune complexes. We did not observe any differences in overall B, CD4 T, or CD8 T cell levels (Fig. 3M–O). The level of GC B cells was slightly higher in the mesenteric lymph nodes (MesLN) of non-injected controls, compared to chimeras receiving the 564Igi antibody (Fig. 3P). There were no differences in GC B cell levels in inguinal lymph nodes (IngLN) and, most importantly, the spleen (Fig. 3P). Furthermore, plasmablast/plasma cell frequencies were comparable between the groups (Fig. 3Q), despite robust levels of idiotype antibodies in injected mice as compared to controls (Fig. 3R).

Taken together, these findings supported that the presence of the autoantibody, even under permissive conditions, is insufficient to elicit a break-of-tolerance.

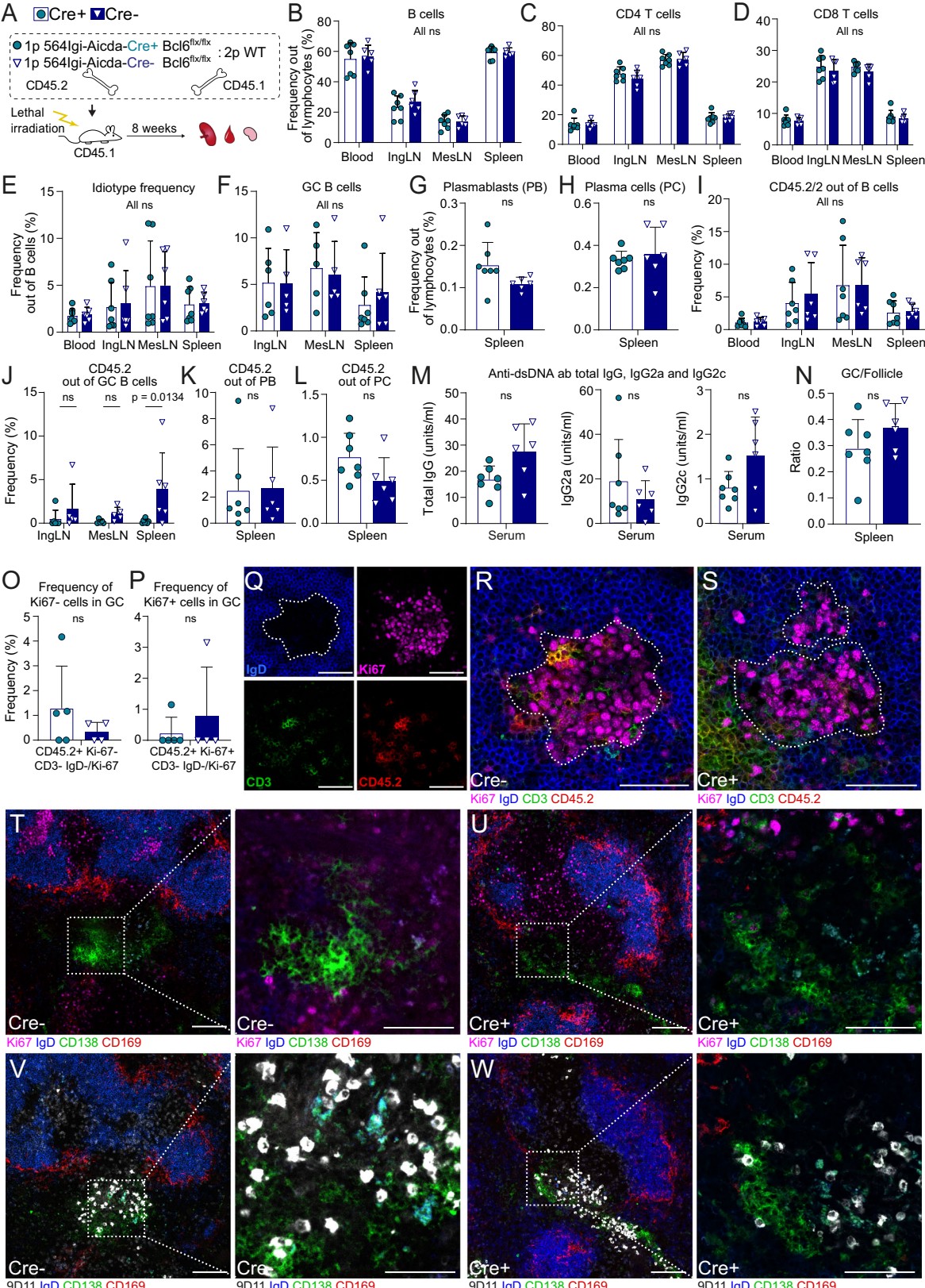

## The initiating clone localizes early to extrafollicular splenic bridging channels and the subsequent epitope spreading depends on MHCII in the WT compartment

The notion that the initiating autoreactive clone did not need to enter GCs, and that it was not the extrafollicular production of auto-antibodies that allowed a break-of-tolerance, combined with the observation of Id+ cells in EFSBCs (Fig. 2T-W), raised a third possibility: the 564Igi B cells could directly instruct a break-of-tolerance in the T-cell compartment, which would subsequently lead to the T-dependent recruitment of proto-autoreactive WT derived B cells. A previous study demonstrated that extrafollicular B-cell activation by dendritic cell-inhibitory receptor 2 (DCIR2) positive marginal zone

**Fig. 2 | The initiating autoreactive clone does not need to enter GCs to break tolerance. A** Experimental setup. 1p = 1 part, 2p = 2 parts. Frequencies of B cells **B**, CD4 **C**, CD8 **D** T cells, and idiotype B cells **E** in blood, spleen, inguinal and mesenteric lymph nodes (IngLN and MesLN, respectively) of Aicda-Cre+ and Aicda-Cre- Bcl6$^{flx/flx}$ 564Igi mixed chimeras. Frequencies of GC B cells **F**, plasmablasts **G**, and plasma cells **H** in secondary lymphoid tissues. Frequencies of CD45.2/2 out of B cells **I**, GC B cells **J**, plasmablasts **K**, and plasma cells **L**. Total IgG (left), IgG2a (middle) and IgG2c (right) anti-dsDNA antibodies in sera **M**. **N** Frequency of GCs per follicle. Frequency of CD45.2$^+$CD3$^-$Ki67$^-$ **O** versus CD45.2$^+$CD3$^-$Ki67$^+$ **P** GC B cells within the GCs (as determined by IgD border). Representative micrograph showing splenic GC of Cre- chimera in split channel **Q** and 4-channel overlay **R**. Broken white line indicates GC border, based on IgD exclusion zone (top left corner). **S** Representative micrograph showing splenic GC of Cre+ chimera in 4-channel

overlay. **T, V** Representative micrographs showing splenic bridging channel in two serial sections of Cre- chimera (left) with higher magnification views of the region of interest indicated by the white boxes (right). **U, W** Representative micrographs showing splenic bridging channel in two serial sections of Cre+ chimera (left) with higher magnification views of the region of interest indicated by the white boxes (right). Data in **B–N** represent two experiments with total n = 7 (Cre + ) and 6 (Cre-) mice, **O–S** represent one experiment with n = 5 (Cre + ) and 4 (Cre-) mice, micrographs in **T–W** represent two mice in each group. Bars and error bars signify mean ± SD in all panels. Two-way ANOVA with Šidák's post-test was used for statistical comparisons in **B–F** and **I–J**, and unpaired, two-tailed t-test with Welch's correction in **G, H**, and **K–P**. ns = p > 0.05. For micrographs, color intensities were adjusted uniformly for visual clarity. Scale bars are 50 μm in **Q–S** and righthand panels in **T–W**, and 100 μm in lefthand panels of **T–W**.

(MZ)–associated CD8α- dendritic cells can drive T cell–dependent antibody responses[36]. In that study, antigen-specific B cells primed by DCIR2+ dendritic cells were found to accumulate within EFSBCs.

To test this possibility, we set up mixed chimeras with one part 564Igi driver BM, one part WT BM, and one part MHC haplosufficient (null/b) or MHC insufficient BM (null/null) (Supplementary Fig. 2A). Four weeks post reconstitution we assayed the inclusion of WT derived cells into the response and evaluated histologically the EFSBC. Overall B-cell and GC B-cell levels were comparable between the two groups (Supplementary Fig. 2B, C), as was the frequency of Id+ cells, plasmablasts and plasma cells (Supplementary Fig. 2D, F and G). Among GC B cells, CD45.2 cells, which could be derived either from the 564Igi BM or the null/b or null/null BM, were present at normal levels in the MHCII$^{null/b}$ group, whereas they were nearly excluded in the MHCII$^{null/null}$ group (Supplementary Fig. 2E). Analysis of I-Ab expression levels among CD45.1 vs. CD45.2 cells of the B-cell compartment confirmed halving of MHCII expression and near-complete ablation in the latter, in null/b and null/null chimeras, respectively (Supplementary Fig. 2H). However, within the splenic GC B-cell compartment of null/null chimeras, the few CD45.2 cells present were I-Ab sufficient, supporting the notion that these represented 564Igi-derived GC B cells (Supplementary Fig. 2I). This demonstrated that inclusion of WT B cells into the GC response required MHCII expression.

Histological evaluation revealed that a robust T-cell zone (CD8, red) had formed around the periarteriolar lymphoid sheath (PALS) by this 4-week time point, surrounded by follicles (B220, white), bounded by a marginal zone (broken line based on CD169 signal) (Supplementary Fig. 3A, B). DCIR2+ DCs (33D1, blue) were seen in the red pulp, and invading the border of the white pulp, where they were juxtaposed with CD138 (Syndecan-1) positive cells (arrowheads). The same anatomical region harbored CD4+ cells and DCIR2+ signal colocalized with CD11c (Supplementary Fig. 3C–E). This region additionally harbored idiotype bright cells (9D11), with an appearance like that of CD138+ plasmablasts (Supplementary Fig. 3F). Co-staining for CD138 and idiotype confirmed this supposition (Supplementary Fig. 3G). The juxtaposition of Id+ cells, CD4 positive cells and DCIR2+ cells was confirmed by additional stainings (Supplementary Fig. 3H–O).

Taken together, this supported early localization of initiating idiotype positive B cells in the EFSBCs and demonstrated dependence of the WT compartment on MHCII.

## Epitope spreading is compartmentalized by MHC haplotype

The MHC locus is highly polymorphic, but in inbred lines of mice, discrete haplotypes of MHC have been fixed. The two classical class II antigens produced by the MHC (*H2*) complex in mice, called I-A and I-E, are each composed of an α– and a β–chain. In many inbred mouse lines, the *I-Eα* gene is non-functional[37]. Thus, MHC$^{b/b}$ strains, such as C57BL/6, express two copies of I-A$^b$, whereas I-E is not expressed. Conversely, MHC$^{d/d}$ strains, such as BALB/cBy, express two copies of I-A$^d$ and two copies of I-E$^d$. Numerous MHC congenic strains have been generated by introgression of particular MHC haplotypes onto specific

backgrounds, enabling the study of MHC function. For example, B6.C-*H2$^d$*/bByJ carry the BALB/cBy *H2* complex (i.e., MHC$^{d/d}$) on C57BL/6 background. For simplicity, when referring to the complete MHC haplotype, e.g., the b haplotype, we use MHC$^{b/b}$, when referring to the MHCII locus, MHCII$^{b/b}$, and for the expressed protein hereof, I-A$^{b/b}$.

To explore the hypothesis that the first autoreactive clone broke tolerance among WT clones by instructing autoreactive T cells, we set up chimeras in which the 564Igi driver compartment and one WT compartment were separated from a second WT compartment by an MHC barrier. Lethally irradiated MHC$^{b/d}$ recipients were reconstituted with BM from 564Igi MHC$^{b/b}$, WT MHC$^{b/b}$ and WT MHC$^{d/d}$ donors. In this scenario, the thymic epithelium of the MHC$^{b/d}$ recipients should support education of both MHC$^b$- and MHC$^d$-restricted T cells, and we hypothesized that 564Igi cells would break tolerance and initiate GCs supported by I-A$^b$-restricted T cells. If true, these would be predicted to subsequently support inclusion of B cells from the WT MHC$^{b/b}$ compartment, but not the WT MHC$^{d/d}$ compartment. As controls, we set up chimeras with MHC$^{b/d}$ BM in lieu of the MHC$^{d/d}$ compartment (Fig. 4A and Supplementary Fig. 4A). The two WT compartments were also distinguishable based on their CD45 isotype (Supplementary Fig. 4A).

Donor marrow was depleted of T and NK cells using a cocktail of antibodies, in order to prevent graft-versus-graft and graft-versus-host reactivity, and recipients were lethally irradiated to facilitate engraftment and induce graft tolerance[38]. At 6–8 weeks post reconstitution, we bled the d/d and b/d chimeras and verified that they had normal frequencies of B cells, CD4 and CD8 T cells (Supplementary Fig. 4B, E, respectively), a balanced representation of I-A$^b$ and I-A$^d$ compartments, as well as MHC class I H2-K$^b$ and K$^d$ compartments (Supplementary Fig. 4C, F, respectively), and displayed circulating Id+ B cells (Supplementary Fig. 4D, G, respectively). The chimeras were subsequently euthanized and IngLN, MesLN and spleen were harvested. The two groups had comparable levels of B cells (Fig. 4B), CD4 T cells (Supplementary Fig. 4H), CD8 T cells (Supplementary Fig. 4I), plasmablasts (Supplementary Fig. 4J), and plasma cells (Supplementary Fig. 4K), although CD8 T-cell levels were slightly lower in the LNs of MHC$^{b/d}$ chimeras. All chimeras displayed circulating Id+ B cells to some degree, although levels were slightly lower in LNs in the d/d group (Fig. 4C). They also presented with robust levels of anti-dsDNA antibodies (Fig. 4D) as a hallmark of the autoimmune phenotype. Direct comparison of cell distributions and I-A haplotype frequencies in the blood of d/d chimeras 7–8 weeks post reconstitution and in blood at the time of euthanasia, from 10 and up to 16 weeks post reconstitution, demonstrated stable chimerism (Supplementary Fig. 4B, C).

Having established that the two groups of chimeras were overall similar and presented with the expected autoimmune phenotype, we asked to what extent the competitor MHC$^{b/b}$ cells and the compartments of interest (MHC$^{d/d}$ or MHC$^{b/d}$) were represented in the total B-cell and GC B-cell repertoire. In the MHC$^{d/d}$ chimeras, the I-A$^{d/d}$ compartment dominated the B-cell population across lymphoid tissues, at roughly 50–75% of the total, with the remainder chiefly constituted by I-A$^{b/b}$ cells, and a small residual of I-A$^{b/d}$ cells from the

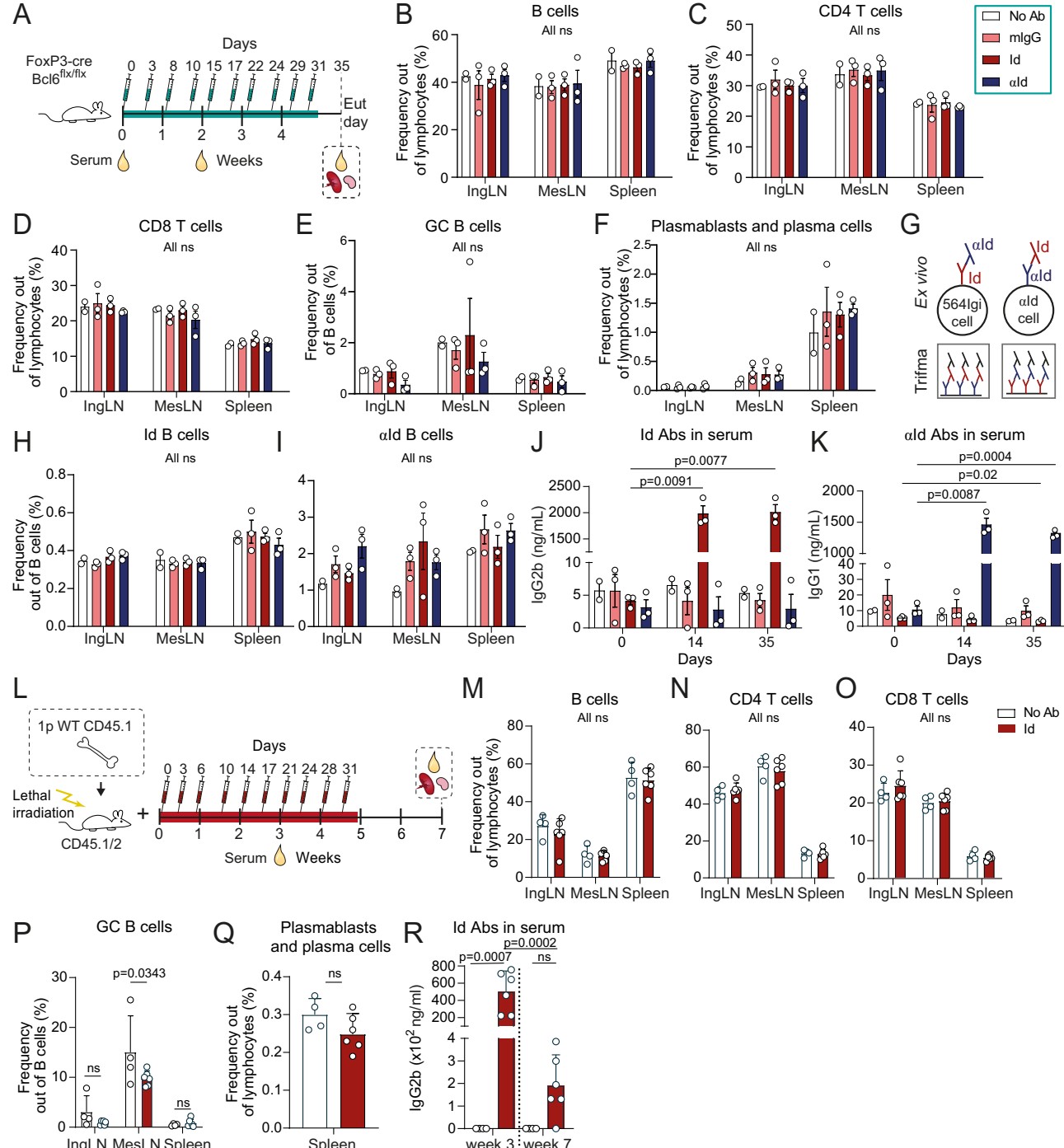

**Fig. 3 | Autoantibody derived from the initiating clone is insufficient to break tolerance, even under permissive conditions. A** Experimental setup to evaluate the potential of the autoreactive antibody from the initiating clone (Idiotype, Id) to drive a break in tolerance in a setting deficient in T follicular regulatory cells, compared to no antibody (not injected), a pool of total normal murine IgG (mIgG), or the anti-idiotype (αId). Frequencies of B cells **B**, CD4 **C**, CD8 **D** T cells, GC B cells **E** and plasmablasts/plasma cells **F** in IngLN, MesLN and spleen across the four groups. **G** Schematic representation of assays and nomenclature for identification of Id+ (564Igi) B cells (top left), anti-idiotypic B cells (top right), idiotype antibodies (564 C11) in sera (bottom left), and anti-idiotypic antibodies (9D11) in sera (bottom right). **H** Idiotype B cell frequency out of total B cells across IngLN, MesLN and spleen in the four groups. **I** Anti-Idiotype B cell frequency out of total B cells across IngLN, MesLN and spleen in the four groups. Idiotype **J** and

anti-idiotype **K** antibodies in sera at 0, 14, and 35 days. **L** Experimental setup to evaluate the potential of the autoreactive antibody from the initiating clone (Idiotype, Id) to drive a break in tolerance during bone marrow reconstitution, compared to no antibody (not injected). 1p = 1 part. Frequencies of B cells **M**, CD4 **N**, CD8 **O** T cells, GC B cells **P**, and plasmablasts/plasma cells **Q** in IngLN, MesLN and spleen across the two groups. **R** Idiotype antibodies (564 C11) in sera of mice presented in panels M-Q. For **A**–**K**, n = 3 mice for mIgG, Id and αId, and n = 2 mice for No Ab. For **L**–**R**, n = 4 (No Ab) and 6 (Id) mice. Bars and error bars signify mean ±SD in all panels. Two-way ANOVA with Tukey's post-test was used for comparisons of data in panels B-F and H-K. Two-way ANOVA with Šidák's post-test was used for **M**–**P** and **R**, and unpaired, two-tailed t-test with Welch's correction was used for statistical comparisons of data in **Q**. ns = p > 0.05.

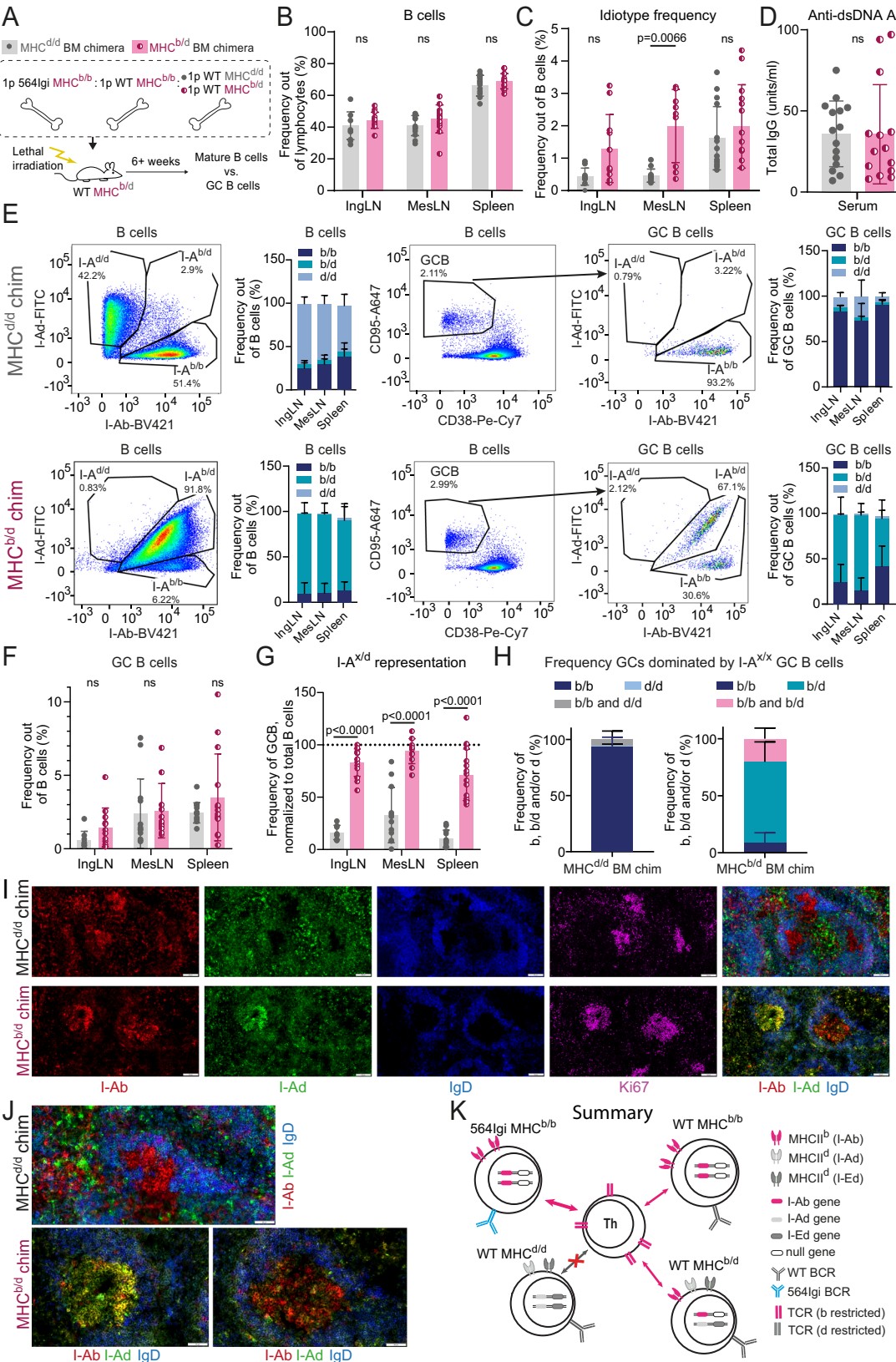

recipient compartment (Fig. 4E, top row). However, the GC B-cell compartment of these chimeras was almost exclusively populated by I-A[b/b] cells. Of note, this was likely not a consequence of a large con- tribution from the I-A[b/b] 564Igi compartment, because 564Igi cells have been found to be almost completely excluded from GCs after 6 weeks[27], and indeed the idiotype frequency was low (Fig. 4C).

Therefore, the I-A[b/b] cells observed in GCs were likely derived from the WT MHC[b/b] compartment. However, to more definitively assess this, we leveraged CD45.1/2 as an independent congenic marker for the WT MHC[b/b] compartment (Supplementary Fig. 4A), which confirmed their fractional representation among B220 (Supplementary Fig. 4L), CD4 T (Supplementary Fig. 4M), and CD8 T (Supplementary Fig. 4N), but

**Fig. 4 | The first autoreactive clone can only break tolerance within an MHC congenic compartment. A** Experimental setup. 1p = 1 part. Frequencies of B cells **B** and idiotype positive B cells **C** 6–8 weeks post reconstitution, in inguinal and mesenteric lymph nodes (IngLN and MesLN), and spleen. **D** Levels of anti-dsDNA in serum 7–16 weeks post reconstitution. **E** For MHC$^{d/d}$ (top) and MHC$^{b/d}$ (bottom) chimeras, representative bivariate plots with gating for I-A$^{b/b}$, I-A$^{d/d}$ and I-A$^{b/d}$ positive cells within the total B-cell population (left); a summary graph of these data (second from left); a representative bivariate plot showing gating for the germinal center (GC) B-cell population (middle), and I-A$^{b/b}$, I-A$^{d/d}$ and I-A$^{b/d}$ gates within the GC B-cell population (second from right); followed by a summary graph of these data (right). **F** Frequencies of GC B cells across IngLN, MesLN and spleen. **G** Representation of I-A$^{d/d}$ and I-A$^{b/d}$ cells within GCs relative to representation in mature B-cell compartment, in direct competition with I-A$^{b/b}$ cells in MHC$^{d/d}$ and MHC$^{b/d}$ chimeras, respectively. Dashed line indicates 1:1

representation. **H** Frequency of individual GCs dominated by I-A$^{b/b}$, I-A$^{d/d}$ cells or both in MHC$^{d/d}$ chimeras (left), or dominated by I-A$^{b/b}$, I-A$^{b/d}$ or both in MHC$^{b/d}$ chimeras (right). Quantification of 14-53 GCs/mouse in 10 (MHC$^{d/d}$) and 3 (MHC$^{b/d}$) chimeras respectively, for a total of 487 GCs in spleen. **I** Representative single-channel images and overlay of I-Ab, I-Ad, and IgD (right) of GCs in spleen of an MHC$^{d/d}$ (top) and an MHC$^{b/d}$ chimera (bottom). **J** Zoomed-in representative images of splenic GCs of an MHC$^{d/d}$ (top) and an MHC$^{b/d}$ chimera (bottom). Overlays colored as in **I**. **K** Schematic interpretation of data. Panels **B**–**G** represent data from $n = 15$ (MHC$^{d/d}$) and 14 (MHC$^{b/d}$) chimeras. Bars and error bars represent mean ± SD. Statistical comparison using two-way ANOVA with Šidák's post-test in **B**, **C**, **F**, and **G**, and unpaired, two-tailed t-test with Welch's correction in D. ns = $p > 0.05$. Images in **I**–**J** are crop-outs of tile scans of spleen sections. Color intensities were adjusted uniformly for visual clarity. Scale bar represents 100 μm **I** and 50 μm **J**.

near-dominance among GC B cells (Supplementary Fig. 4O), plasma-blasts and plasma cells (Supplementary Fig. 4P). In the MHC$^{b/d}$ chimeras, the I-A$^{b/d}$ compartment made up the bulk of the B-cell population, ~85% of the total across lymphoid tissues, with the remainder being I-A$^{b/b}$ cells (Fig. 4E, bottom row). Despite their numeric inferiority, I-A$^{b/b}$ cells were still well-represented in GCs, but I-A$^{b/d}$ cells also contributed to a great extent. Again, the relative representation of the WT MHC$^{b/b}$ compartment was confirmed across populations and tissues using an independent congenic marker profile (CD45.1 or CD45.1/2) (Supplementary Fig. 4Q–U). Relatively robust GC B-cell levels were present in the chimeras, in the three lymphoid tissues examined, and to a comparable extent in the two groups (Fig. 4F). We quantified the frequency of the compartment of interest (I-A$^{d/d}$ or I-A$^{b/d}$) within the GC B-cell gate, relative to their frequency within the total B-cell pool, as a measure of their competitive representation in GCs. Whereas I-A$^{b/d}$ cells were nearly represented 1:1, I-A$^{d/d}$ cells were virtually excluded from GCs in the spleen and IngLN, and to a slightly lesser extent from GCs in the MesLN (Fig. 4G). The differential behavior of MesLN GCs was likely explained by their involvement in gut microbiota-directed responses.

To understand how the differential inclusion of I-A$^{b/d}$ and I-A$^{d/d}$ cells played out at an individual GC level, we performed imaging of spleen sections and quantified the frequency of GCs dominated by I-A$^{b/b}$, I-A$^{d/d}$ or both for MHC$^{d/d}$ chimeras and GCs dominated by I-A$^{b/b}$, I-A$^{b/d}$ or both for MHC$^{b/d}$ chimeras (Fig. 4H–J). This confirmed the dominance of I-A$^{b/b}$ and exclusion of I-A$^{d/d}$ cells in GCs of MHC$^{d/d}$ chimeras, and the dominance of I-A$^{b/d}$ cells in GCs of MHC$^{b/d}$ chimeras. We also observed rare, mixed I-A$^{b/b}$ and I-A$^{d/d}$ positive GCs in the MHC$^{d/d}$ chimeras, whereas about a fifth of the GCs in the MHC$^{b/d}$ chimeras had I-A$^{b/b}$ and I-A$^{b/d}$ B cells co-existing in the same GC. The occasional observation of small numbers of infiltrating I-A$^{d/d}$ cells in I-A$^{b/b}$-dominated GCs in the MHC$^{d/d}$ chimeras (Supplementary Fig. 4V) was likely a reflection of the previously reported open and accessible nature of the GC light zone[39].

Taken together, this indicated that I-A$^{b/d}$ cells were only slightly disadvantaged in autoreactive GCs when competing with I-A$^{b/b}$ cells in a response initiated by 564Igi I-A$^{b/b}$ drivers. I-A$^{d/d}$ cells on the other hand were dramatically disadvantaged in GCs of the spleen and IngLNs, and slightly less so in GCs of the MesLN, which are in part directed at the commensal microbiota. The most parsimonious explanation is that the 564Igi B cells break tolerance of I-A$^{b}$-restricted T cells, which can in turn recruit proto-autoreactive I-A$^{b/b}$ and I-A$^{b/d}$ B cells to the GC response, whereas I-A$^{d/d}$ cells are excluded (Fig. 4K).

### The dominance of I-A$^d$ cells in the B-cell repertoire of mixed chimeras is intrinsic and independent of the B-cell haplotype initiating autoreactivity

It was somewhat surprising that I-A$^{b/d}$ and I-A$^{d/d}$ cells constituted upwards of 80% of the total B-cell compartment of mixed chimeras (Fig. 4E and Supplementary Fig. 4L–U). We considered that this could indicate a negative selection effect on I-A$^{b/b}$ cells, due to the role of

564Igi I-A$^{b/b}$ cells in instigating the autoimmune response. To examine whether this could be the case, we set up non-autoimmune chimeras, again using lethally irradiated MHC$^{b/d}$ recipients, but reconstituting with one part MHC$^{d/d}$ BM and either one or two parts MHC$^{b/b}$ BM (Supplementary Fig. 5A). We assessed the overall levels of major cell types (Supplementary Fig. 5B) and distribution of the MHC$^{b}$- and MHC$^{d}$-linked allotypes of the ubiquitously expressed MHC class I molecule H2-K, in addition to I-A, as shown for representative spleen samples of an MHC$^{d/d}$ control, a 1:1 mixed chimera, and an MHC$^{b/b}$ control (Supplementary Fig. 5C). Expression of I-A$^d$ and I-A$^b$ on the cell surface was assessed by determining median fluorescence intensity (MFI) for either haplotype on B cells (Supplementary Fig. 5D). This revealed I-A isotype expression levels commensurate with the underlying haplotype of the cell compartments in question. In the blood (Supplementary Fig. 5E), IngLN (Supplementary Fig. 5F), MesLN (Supplementary Fig. 5G), and spleen (Supplementary Fig. 5H), the H2-K$^d$ cells made up approximately 80% of B220, CD4 T and CD8 T cells in 1:1 chimeras, and 50% in 1:2 chimeras, and the same was true when using I–A for haplotype assignment among B cells. The picture was comparable among the CD11c-positive dendritic cells (Supplementary Fig. 5E–H).

Taken together, this demonstrated that the over-representation of I-A$^{d/d}$ B cells in mixed chimeras observed in the previous setups was likely an intrinsic effect, independent of the autoimmune environment. Furthermore, this was not restricted to the B-cell compartment, but extended to other cell compartments expressing (dendritic cells) or not expressing (e.g., CD4 T and CD8 T cells) MHC class II. Hence, it was likely an effect of differences in the hematopoietic potential of the different strains. Since the MHC$^{d/d}$ strain is derived by introgression of the $H2^d$ locus from BALB/cBy into C57BL/6J, the background of the MHC$^{d/d}$ and MHC$^{b/b}$ strains is essentially identical, save for the MHC locus. Therefore, the basis of this intrinsic difference in hematopoietic potential resides within or is directly linked to the MHC locus.

### A bridging B-cell compartment enables epitope spreading across an MHC barrier

Having established that MHC$^{d/d}$ cells were overall not behaving fundamentally differently in the BM chimeras and noting that I-A$^{b/d}$ cells were well capable of participating in the response, it seemed plausible that the reason for exclusion of I-A$^{d/d}$ cells from the autoreactive response in MHC$^{d/d}$ chimeras (Fig. 4G) was indeed due to the MHC barrier separating them from the eliciting MHC$^{b/b}$ compartment. If this were true, we hypothesized that the epitope spreading could potentially be bridged by B-cell antigen presentation across MHC haplotypes. To test this, we set up mixed BM chimeras using 564Igi MHC$^{b/b}$ BM as an autoreactive driver, WT MHC$^{b/d}$ BM and WT MHC$^{d/d}$ BM as potential epitope spreading compartments, in lethally irradiated MHC$^{b/d}$ recipients (Fig. 5A and Supplementary Fig. 6A). At 6 weeks post reconstitution, we bled the mice and confirmed normal levels of B

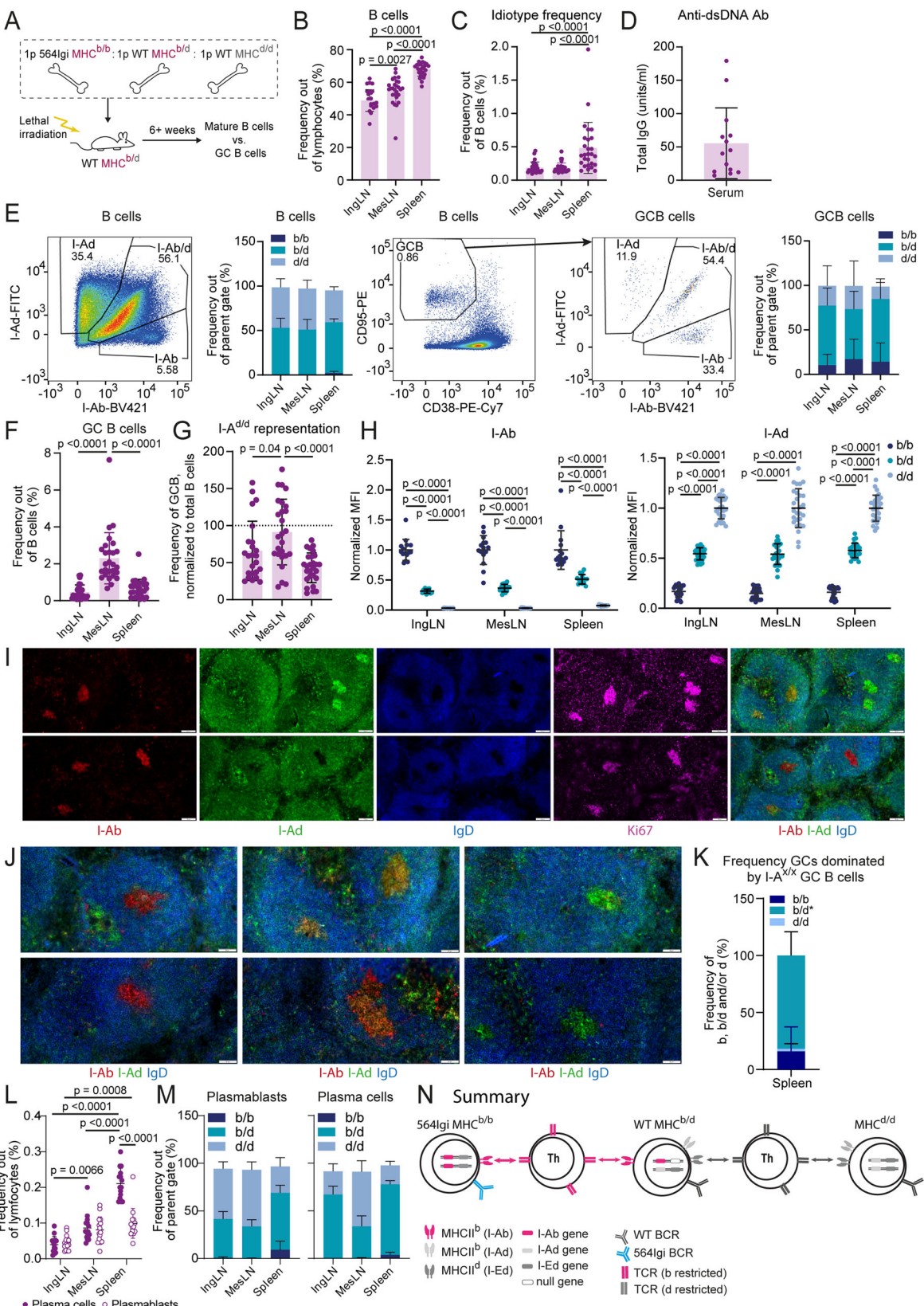

cells, CD4 and CD8 T cells (Supplementary Fig. 6B). In the B-cell compartment, MHCᵇ/ᵇ cells were virtually absent, in agreement with follicular exclusion of the 564Igi cells, and accordingly, MHCᵇ/ᵈ and MHCᵈ/ᵈ cells made up approximately 50% each (Supplementary Fig. 6C). In the CD4 and CD8 T-cell compartments, 564Igi BM derived cells were present, albeit to a lower extent than MHCᵇ/ᵈ and MHCᵈ/ᵈ

derived cells (Supplementary Fig. 6D). Commensurate with their follicular exclusion, there was a low, but detectable level of Id+ cells (Supplementary Fig. 6E). As a further confirmation of haplotype assignment, we measured the MFI of I-Aᵇ and I-Aᵈ on I-Aᵇ/ᵇ, I-Aᵇ/ᵈ and I-Aᵈ/ᵈ cells and observed a titration congruent with MHC haplotype (Supplementary Fig. 6F).

**Fig. 5 | A bridging B cell compartment enables epitope spreading across an MHC barrier. A** Experimental setup. 1p = 1 part. B cells **B** and idiotype+ B cells **C** in inguinal and mesenteric lymph nodes (IngLN and MesLN), and spleen. **D** Anti-dsDNA antibodies in sera. **E** Representative bivariate plot showing gating for I-A$^{b/b}$, I-A$^{d/d}$, and I-A$^{b/d}$ cells among B cells (left); a summary graph of these data (second from left); a representative bivariate plot showing gating for GC B cells (middle), and I-A$^{b/b}$, I-A$^{d/d}$ and I-A$^{b/d}$ gates among GC B cells (second from right); followed by a summary graph of these data (right). **F** GC B-cell frequencies across lymphoid organs. **G** Representation of I-A$^d$ cells within GCs relative to their representation in the mature B-cell compartment, across IngLN, MesLN, and spleen. The dashed line through 100% indicates 1:1 representation. **H** Median fluorescence intensity (MFI) for I-A$^b$ (left) and I-A$^d$ (right), within the I-A$^{b/b}$, I-A$^{b/d}$, and I-A$^{d/d}$ B-cell compartments of the chimeras, across IngLN, MesLN, and spleen. **I** Representative images of GCs in spleen of a bridge chimera. **J** Zoomed-in representative images of GCs in the spleens of bridge chimeras. **K** Frequencies of individual GCs dominated by I-A$^{b/b}$, I-A$^{d/d}$, or both. Quantification based on 6–50 GCs/mouse in 13 mice for a total of 326 GCs. *This population can consist of I-A$^{b/d}$ positive cells only or any combination of I-A$^{b/b}$, I-A$^{b/d}$ and/or I-A$^{d/d}$. **L** Plasma cell and plasmablast frequencies across IngLN, MesLN, and spleen. **M** I-A haplotype distribution among plasmablasts (left) and plasma cells (right) across IngLN, MesLN, and spleen. **N** Schematic interpretation of results. Throughout panels, error bars represent mean ± SD, and n = 27 mice total from two independent experiments. In **B**, **C**, **F**, and **G**, Kruskal–Wallis test with Dunn's post-test was used, in **H**, two-way ANOVA with Tukey's post-test, and in **L**, one-way ANOVA with Tukey's post-test. Images in **I**–**J** are crop-outs of tile scans of spleen sections. Color intensities were adjusted uniformly for visual clarity. The data were procured 7–9 weeks post reconstitution. Scale bars represent 100 μm **I** and 50 μm **J**.

Chimeras were euthanized and we observed normal B-cell frequencies across IngLN, MesLN, and spleen (Fig. 5B), appreciable Id+ cell levels in the spleen (Fig. 5C), and anti-dsDNA antibodies in the serum (Fig. 5D). We also verified normal levels of major cell subsets across lymphoid tissues (Supplementary Fig. 6G, H) and chimerism in lymphoid tissues reflective of that observed in blood (Supplementary Fig. 6I–M). Lymphocyte subset frequencies and degree of chimerism were comparable between blood sampling at 6 weeks and at the time of analysis at 7 or 9 weeks post reconstitution, confirming the chimerism to be stable over time (Supplementary Fig. 6A–E).

We hypothesized that in this scenario, MHC$^b$-restricted T-cell tolerance would be broken to allow the inclusion of MHC$^{b/d}$ B cells in GC responses, and these in turn should be able to support break-of-tolerance of MHC$^d$-restricted T cells, hence serving a bridging function for subsequent inclusion of MHC$^{d/d}$ B cells. To evaluate this, we determined by flow cytometry the relative frequencies of I-A$^{b/d}$ and I-A$^{d/d}$ positive cells, in the total B-cell compartment versus the GC B-cell compartment, across IngLN, MesLN, and spleen (Fig. 5E). We verified an appreciable GC B-cell frequency (Fig. 5F), then determined the representation of I-A$^{d/d}$ cells in the GC B-cell population relative to that of the total B-cell pool (Fig. 5G). The average representation of MHC$^{d/d}$ cells in GCs of the MesLN was ~1:1, serving as an internal control that MHC$^{d/d}$ cells were intrinsically equally capable of partaking in the mixed GC responses towards the gut microbiome. In GCs of spleen and IngLN in these bridge chimeras, I-A$^{d/d}$ cells were present at an appreciable level (Fig. 5G), in stark contrast to the near-complete exclusion observed in our MHC$^{d/d}$ no bridge chimeras (Fig. 4G). We verified across the lymphoid tissues that the I-A$^b$ and I-A$^d$ expression levels corresponded to the haplotype assignment for both B cells (Fig. 5H) and GC B cells (Supplementary Fig. 6N).

To corroborate our findings at an individual GC level, we performed imaging of spleen sections and quantified the frequency of GCs dominated by I-A$^{b/b}$, I-A$^{d/d}$ or containing both and/or I-A$^{b/d}$ cells (Fig. 5I–K). This confirmed the presence of GCs dominated by only I-A$^{d/d}$ or I-A$^{b/b}$ cells, but revealed a preponderance of GCs with an I-A$^{b/d}$ phenotype (Fig. 5K), which could be a mix of I-A$^{b/b}$ and I-A$^{d/d}$ expressing cells, pure I-A$^{b/d}$ expressing cells, or most often a mix of two or all three cell types (these various possibilities could often, but not always, be discriminated).

Finally, to assess the downstream functional impact of the GC participation of the three haplotype subsets, we assessed the output of plasmablasts and plasma cells across the lymphoid tissues (Fig. 5L) and determined the haplotype distribution among these (Fig. 5M). This revealed that I-A$^{d/d}$ cells contributed robustly to the plasmablast and plasma cell compartments of the MesLN. They contributed appreciably, albeit less robustly, to the IngLN and splenic plasmablast and plasma cell compartments, to an extent that was overall in agreement with their representation in the GCs.

Taken together, these findings suggested that in bridge chimeras, I-A$^{d/d}$ B cells were equally able to participate in the foreign antigen-directed GC response of the gut, and equally efficient at generating plasmablast and plasma cell output from this response. They were somewhat disadvantaged in the autoreactive response initiated by 564Igi MHC$^{b/b}$ cells, both in terms of GC participation and plasmablast and plasma cell output, but importantly they were able to partake in the response. This was in stark contrast to the MHC$^{d/d}$ no bridge chimeras, in which I-A$^{d/d}$ cells were virtually excluded from GCs (Supplementary Fig. 6O). Furthermore, there was a direct correlation between the frequency of I-A$^d$ expressing mature B cells and the frequency of I-A$^d$ expressing GC B cells in the bridge chimeras in IngLN, MesLN, and spleen, whereas no such correlation was observed in the no bridge chimeras (Supplementary Fig. 6P). The observation that I-A$^{d/d}$ cells were present in splenic and cutaneous LN GCs, in 564Igi MHC$^{b/b}$-driven autoreactive chimeras, strongly suggested that the epitope spreading process was able to cross the MHC barrier in this model (Fig. 5N).

## Repertoire sequencing and autoantigen arrays confirm respective autoreactive I-A$^{d/d}$ cell exclusion and inclusion in GCs of no bridge and bridge chimeras

To better understand the immune landscape of the no bridge and bridge chimeras (Fig. 4 + Fig. 6A and Fig. 5 + Fig. 6B, respectively), we leveraged the BD Rhapsody platform to perform linked single-cell Abseq, immune profiling, and BCR/TCR analyses. We analyzed total splenocytes from two no bridge chimeras, and total splenocytes enriched for GC B cells from a bridge chimera. Approximately 38,000 cells in total were subjected to clustering and 2D representation by tSNE dimensionality reduction (Fig. 6C). This revealed the presence of the major splenic leukocyte subsets, dominated by B cells with a minor population of GC B cells, followed by CD4 and CD8 T cells, as identified by canonical cell subset markers (Supplementary Fig. 7A). We gated out the B cell populations (Supplementary Fig. 7B) and subjected these to re-clustering followed by UMAP representation, further separating out GC B cells (Cluster 7, Fig. 6D, E and Supplementary Fig. 7C). B220+ cells were gated based on AbSeq staining (Fig. 6F), and stratified into I-A$^{b/b}$, I-A$^{b/d}$ and I-A$^{d/d}$ among all (Fig. 6G) or each (Fig. 6H–J) of the samples. This confirmed relative haplotype representation among all B220+ cells in line with that of the parental group of chimeras, as established by flow cytometry (Figs. 4 and 5). To understand the relative inclusion of B cells of each haplotype into the GC response, we compared the haplotype representation among the primary repertoire (naive B cells in clusters 0, 1, 2, 3, i.e, Naive 1–4) and GC B cells (cluster 7) (Fig. 6K). This confirmed our finding that in no bridge chimeras, despite a numerical superiority of MHC$^{d/d}$ cells in the primary repertoire, these cells were by-and-large excluded from GC participation. GCs in these chimeras were instead dominated by b/b cells. Conversely, in the bridged setup, d/d cells were nearly represented 1:1 in GCs relative to the primary repertoire, but b/d, and to a lesser extent b/b cells were also present in GCs (Fig. 6K). Importantly, the dominance of b/b cells in GCs of no bridge chimeras and their presence in GCs of

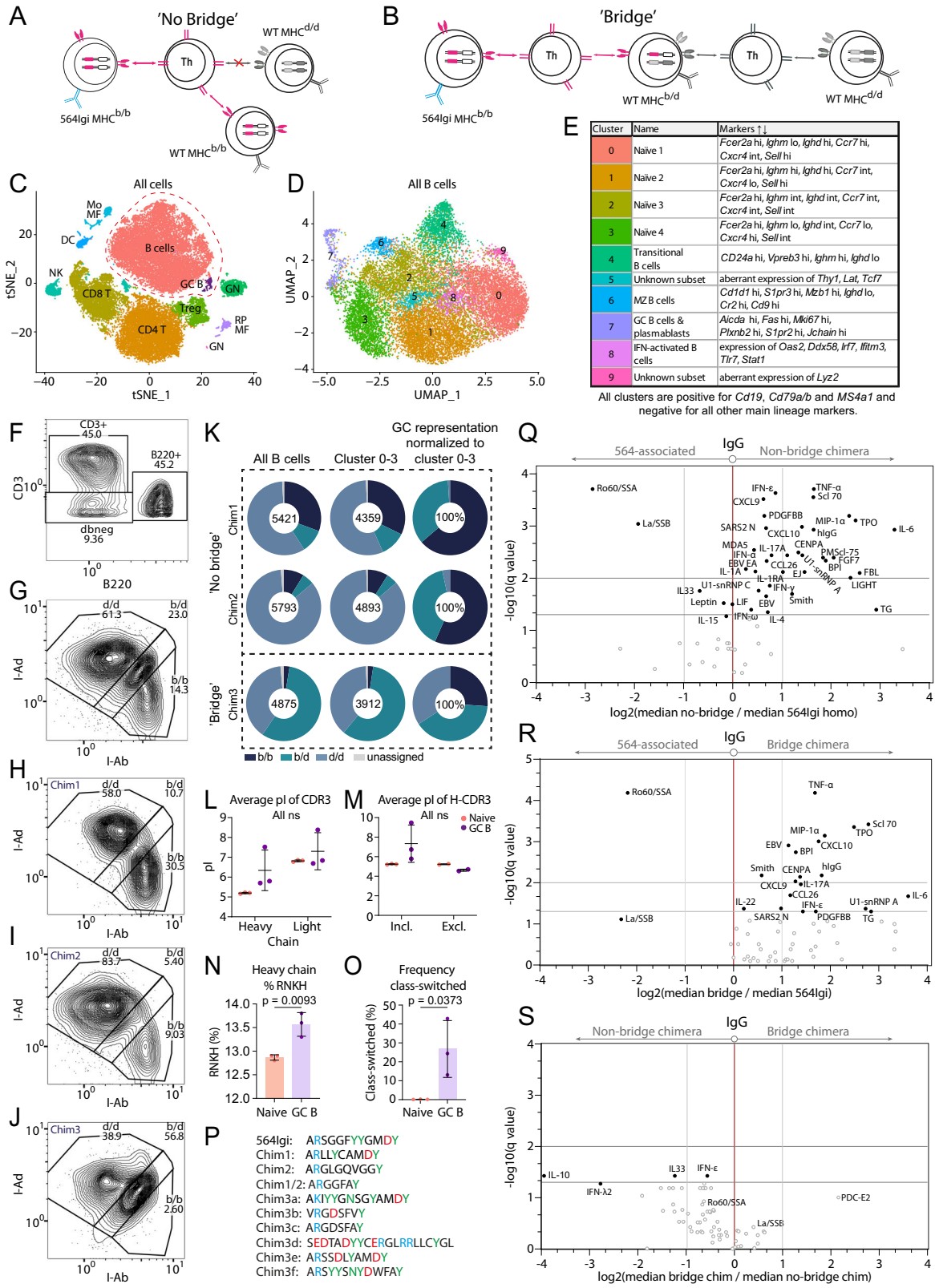

the bridged setup was not caused by a significant contribution of 564Igi-derived cells, because we could identify only 16 cells closely resembling (2 cells) or directly carrying (14 cells) the 564Igi knock-in heavy and light chain out of 16,089 total clones analyzed, and none of these mapped to cluster 7.

The paucity of 564Igi cells was well in line with the previously described exclusion of these cells from the repertoire 6 weeks

following reconstitution[27], and with the low idiotype frequencies observed in our chimeras by flow cytometry (Supplementary Fig. 4D and 6E). To ascertain that the observed GC B cells did indeed represent an autoreactive response, we asked whether they displayed hallmark features of autoreactive B cells. As shown in Fig. 6L, the average iso-electric point (pI) of CDR3s of the heavy and light chain of GCBs was noticeably higher than that of their naive counterparts. Although this

**Fig. 6 | Sequence and array analyses of no bridge and bridge chimeras.** Hypothesized cell interactions for germinal center (GC) entry in no bridge chimeras **A** and bridge chimeras **B**. **C** Global clustering of ~38,000 splenocytes from no bridge (Chimera 1 and 2) and bridge (Chimera 3) chimeras. For the bridge chimera, splenocytes were additionally enriched for GC B cells. **D** UMAP dimensionality reduction of B and GC B-cell populations from **C**. **E** Cluster definitions for **D**. Highly differentially expressed markers are listed. **F** Gating for CD3 + , B220+ and dbneg populations based on Abseq. **G** Stratification of B220+ population into MHC$^{b/b}$ (b/b), MHC$^{b/d}$ (b/d) or MHC$^{d/d}$ (d/d), based on Abseq. As **F** but split for no bridge chimera 1 **H** and 2 **I**, and bridge chimera 3 **J**. **K** MHCII haplotype distribution among total and naive (Clusters 0–3 from **D**) B cells, and among GC B cells (Cluster 7 from **D**) relative to their representation in the primary repertoire, for each of the three chimeras. **L** Average isoelectric point for the agglomerated heavy and light chain CDR3s of naive (peach) and GC (purple) B cells. **M** Average isoelectric point for the agglomerated heavy chain CDR3s of naive (peach) vs. GC (purple) B cells, stratified by whether they are predicted to be included in (I-A$^{b/b}$ of chimeras 1 + 2 and I-A$^{d/d}$ of chimera 3) or excluded from (I-A$^{d/d}$ of chimera 1 + 2) the autoreactive response. **N** Frequency of arginine, asparagine, lysine, and histidine in the agglomerated CDR3s of naive vs. GC B cells. **O** Class-switched frequency among naive vs. GC B cells. **P** CDR3 sequences of the 10 most expanded GC B-cell clones. **Q** Volcano plot of median IgG signal intensities in no bridge chimeras ($n = 14$) vs. homozygous 564Igi mice ($n = 12$). **R** As Q, but bridge chimeras ($n = 16$) vs. homozygous 564Igi mice ($n = 12$). **S** As Q, but no bridge chimeras ($n = 14$) vs. bridge chimeras ($n = 16$). Bars represent mean ± SD. Statistical comparison in **L** and **M** by two-way ANOVA with Šidák's post-test, and in **N** and **O** unpaired, two-tailed t-test, for $n = 3$ mice. ns = $p > 0.05$.

did not achieve statistical significance, it could indicate a preferential inclusion or expansion of B-cell clones with cationic paratopes within GCs. Strikingly, albeit not statistically significant, this appeared to hold true when we stratified for MHC haplotypes predicted to be included in the response, but not for those predicted to be excluded (Fig. 6M). That is, I-A$^{b/b}$ clones in GCs of no bridge chimeras and I-A$^{d/d}$ clones in GCs of the bridge chimera trended towards higher pI than their naive counterparts, whereas the I-A$^{d/d}$ clones in GCs of no bridge chimeras did not. While this verified that GC B cells of haplotype origin predicted to be included in the autoreactive response displayed the stereotypical cationic paratope, it also indicated that a small number of GC clones were not participating in the autoreactive response, or targeted autoantigens with distinct epitope features. However, this was a small fraction of the GC B cells, 2% (3 out of 145) and 27% (15 out of 56), in line with our previous observation in 564Igi mixed chimeras that autoreactivity could be confirmed in 3 out of 4 GCs analyzed[27].

To further corroborate our findings, we asked whether GC clones in the chimeras presented a higher frequency of arginine, asparagine, lysine, and histidine in their CDR3s, as predominance of cationic residues and asparagine has previously been associated with autoantibodies[40,41]. As seen in Fig. 6N, these residues were indeed enriched in CDR3s of GC clones from the chimeras. Furthermore, we observed a significantly increased propensity for class-switching, further supporting their GC quality (Fig. 6O), which was additionally underlined by their expression profile including hallmark GC genes such as *Mki67*, *Aicda*, and *Fas* (CD95) (Supplementary Fig. 7D–F). Notably, these cells also expressed *Tbx21* (T-bet) (Supplementary Fig. 7G), a signature transcription factor of autoreactive B cells associated with a Th1 response and lupus-like autoimmunity[42,43].

We also inspected the qualitative features of the CDR3s of the 10 most expanded GC B-cell clones in the three chimeras (Fig. 6P). The most prevalent was the public clonotype representing the 564Igi knock-in clone, which was found in all three chimeras. Of the remaining 9, one was shared among chimeras 1 and 2, indicating some degree of convergent evolution of the response. Apart from cationic residues, asparagine, and acidic residues, we noted also the presence of significant numbers of tyrosines across the CDR3s, a feature which has previously been associated with hydrogen bonds between these amino acids and DNA[40].

To further evaluate if functional epitope spreading was indeed occurring in the mixed chimeras, we performed high-density autoantigen arrays of sera from 564Igi homozygous mice, which are locked in to express the knock-in receptor, and of sera from our no bridge and bridge chimeras. This revealed epitope spreading away from an initial 564Igi-directed focus on Ro60/SSA and La/SSB to include a breadth of commonly targeted autoantigens in both no bridge (Fig. 6Q) and bridge chimeras (Fig. 6R). The response appeared slightly weaker with somewhat fewer targets in bridge chimeras, potentially a consequence of a delay caused by the requirement for bridging distinct compartments. However, when comparing the two groups of chimeras, only a

couple of targets displayed significantly higher responses in no bridge chimeras (Fig. 6S).

Building on prior methodology[44], we additionally performed an unbiased autoantigen screening based on pulldown of antigens from precleared spleen extracts using Protein G beads loaded with serum autoantibodies, followed by mass spectrometry identification (Fig. 7A). We first verified the method by pulldowns using monoclonal antibody representing the 564Igi knock-in (clone 564Igi C11, IgG2b), the anti-idiotypic antibody (clone 564Igi 9D11, IgG1), or purified polyclonal IgG from wild-type C57BL/6J mice (Fig. 7B and Supplementary Fig. 8A, B). Out of 182 identified targets total, clone C11 was seen to specifically yield around 40 targets, including 60 S and 40 S ribosomal proteins, nucleolin, smD3, U1 small nuclear ribonucleoprotein A, and several other nucleic acid associated antigens; clone 9D11 extracted only 7 discrete hits: three WASH complex subunits, a WAS protein family homolog, two tubulin gamma complex components, and H2-A beta chain; whereas mIgG yielded three hits: ceruloplasmin, transthyretin, and carboxylesterase 1 C (Fig. 7B and Supplementary Fig. 8A, B). Having verified that clone C11 identified hits associated with ribonuclear complexes, we performed similar pulldowns using beads loaded with sera from 5 individual no bridge or 5 individual bridge chimeras, using as controls a pool of serum from 5 564Igi homozygous mice or 6 C57BL/6J mice. Principal component analysis revealed discrete clustering of no bridge and bridge chimeras together, with robust separation of both 564Igi and C57BL/6J samples (Fig. 7C). Heatmapping demonstrated distinct hit profiles for 564Igi and C57BL/6J, the former again including ribosomal and other ribonucleic acid-associated targets, and distinct, discrete hits for individual no bridge and bridge chimeras indicative of epitope spreading (Fig. 7D).

Based on these analyses, we were unable to definitively determine if a given autoantibody originated in a germinal center or an extrafollicular focus. However, the feasibility of this approach demonstrates that (at least a subset of) the autoantibodies have a considerable affinity, indicating that they likely originate from GC responses displaying a higher degree of affinity maturation.

Taken together, these findings supported the notion that the observed response in the chimeras was indeed autoreactive and confirmed the finding that d/d cells were excluded from the autoreactive GC response in non-bridged chimeras but included in a bridged setting. This could be related back to the respective inability or ability (via an intermediary) of d/d cells to communicate with the initiating autoreactive compartment. In summary, this identified a cellular relay system of autoreactivity (Fig. 8).

## Discussion

Here, we followed the trajectory of the break-of-tolerance initiated by a single, autoreactive B-cell clone in a mixed chimera model of autoreactivity resembling that of SLE. We found that two weeks after irradiation and reconstitution with a 1:2 mix of BM carrying a prerearranged autoreactive knock-in B cell receptor and WT BM, there is a high frequency ( > 20%) of cells carrying the knock-in receptor and an

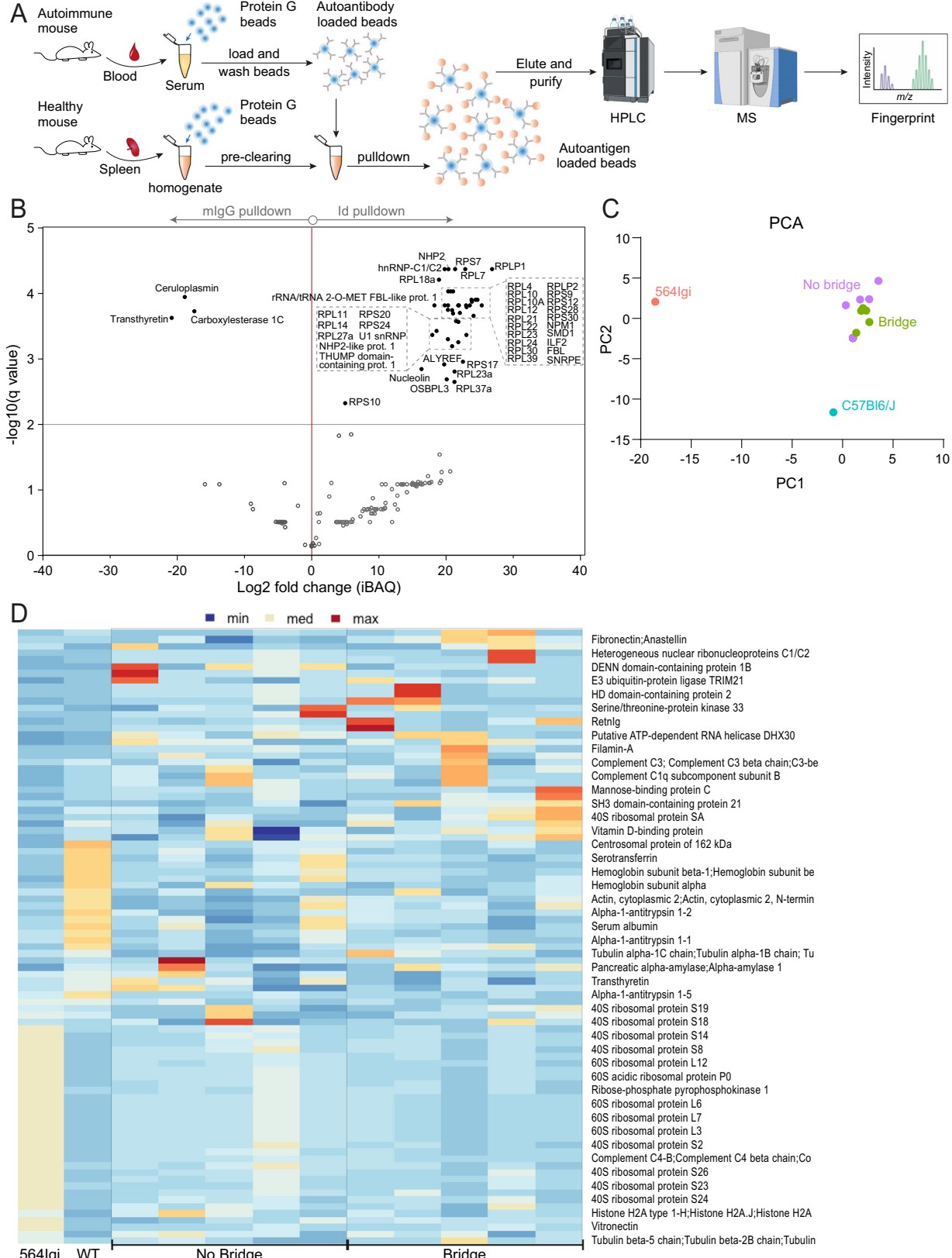

**Fig. 7 | Proteomics analyses of autoantigens. A** Schematic overview of experimental approach. **B** Volcano plot of mean differences in log2[iBAQ values] of clone 564 C11 pulldowns (*n* = 4) and murine serum IgG pulldowns (*n* = 4). The line at -log10(q value) = 2 indicates the significance threshold for paired t-test at *p* = 0.01.

**C** Plot of PC1 and PC2 for principal component analysis of five bridge chimeras, five no bridge chimeras, a pool of five 564Igi homozygous sera, and a pool of six C57BL/6J sera. **D** Heatmap of individual hits for the samples represented in the PCA analysis in **C**. Some graphical elements of **A** were created with BioRender.

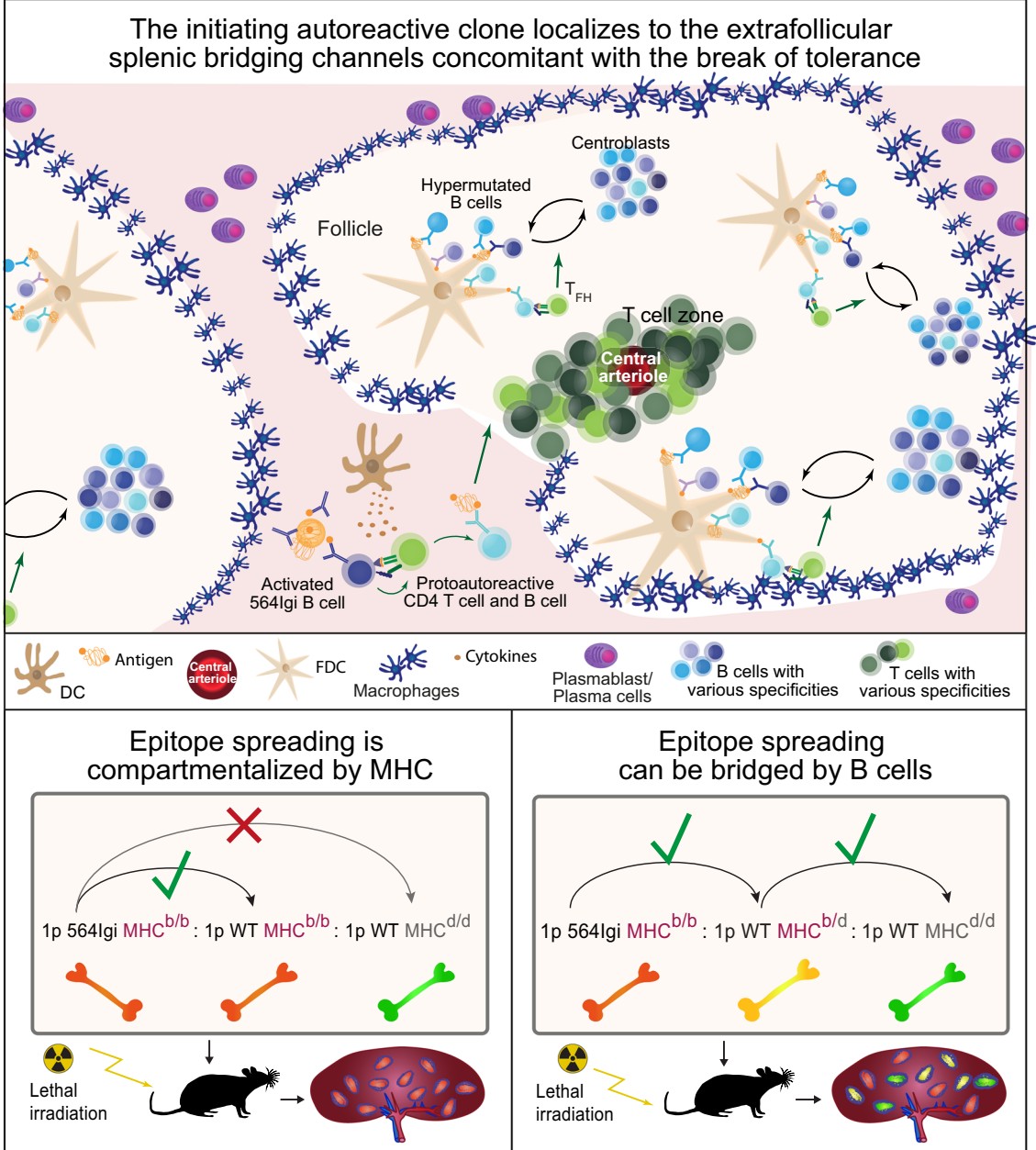

**Fig. 8 | Proposed model and graphical summary of the main finding.** Top, proposed model: the initiating autoreactive clone localizes to the extrafollicular splenic bridging channels, where it may interact with DCIR2+ DCs and elicit a break-of-tolerance in protoautoreactive CD4 T cells, which can subsequently support the formation of autoreactive GCs seeded by wild-type derived B cells. Bottom, summary of the main finding: in mixed chimeras, wild-type B cells only participate in the autoreactive response if they share the MHC haplotype with the initiating clone (left); however, in the presence of a bridging B-cell compartment carrying both MHC haplotypes, these cells are included in the response (right).

abnormally high frequency of B cells in the spleen (Fig. 1). However, no GCs were observed at this time-point. Our interpretation is that following the depletion of B cells by irradiation there is an initial repopulation by 564Igi-derived B cells, which may emerge more rapidly from the BM than their WT counterparts, as they do not need to undergo recombination. The 564Igi B cells are allowed to populate the periphery because of relaxed negative selection, as the negative selection is dependent on the presence of normal competitor cells[45]. As the compartment is filled with normal WT B cells, the negative selection at the transitional stage is reestablished, leading to a drop in naive 564Igi B-cell frequency in the periphery. We surmised that at this point, tolerance had already been broken at the (pre-)GC (potentially T-cell) level, but the GC phenotype did not manifest until sometime between 4 and 6 weeks (Fig. 1). We could not identify any time-point, at which GCs were present, where they were dominated by 564Igi cells, suggesting that 564Igi cells do not directly seed GCs. To test this observation further, we generated 564Igi mice which had a block in the GC pathway, then assayed the ability of BM from these to initiate a break-of-tolerance in the mixed chimera model. This demonstrated more definitively that GC participation of the 564Igi clone is not required for a break-of-tolerance (Fig. 2).

We observed idiotype bright cells in the EFSBCs and red pulp of mixed chimeras (Fig. 2). It has previously been noted that antigen affinity determines plasma cell versus GC fates[46]. As the 564Igi knock-in receptor is already affinity matured, it seemed plausible that antigen-stimulated 564Igi cells would differentiate through the extrafollicular

pathway and rapidly produce autoantibody. Studies have shown that the transfer of serum or autoantibodies from a range of autoimmune diseases can precipitate autoimmune disease in naive mice[47]. To test whether the presence of autoantibody produced by the initiating 564Igi clone was sufficient to break tolerance in a permissive environment, we first treated FoxP3-Cre Bcl6[flx/flx] mice, deficient in T follicular regulatory cells, with the purified antibody. However, this did not elicit a notable response, compared to non-injected controls, total murine IgG injected controls, or controls injected with the anti-idiotype antibody (Fig. 3). This additionally demonstrated that the anti-idiotype antibody did not precipitate an autoimmune response through induction of idiotypic antibodies. The possibility remained, however, that the environment did not adequately recapitulate salient features of the mixed chimera setting. To further approximate this, we repeated the experiment, but this time injecting the autoantibody into WT BM chimeras. This also failed to induce a response comparable to that of the mixed chimeras, confirming the insufficiency of the auto-antibody itself in breaking tolerance (Fig. 3). This suggested an absolute requirement for the presence of the initiating clone itself.

To begin to uncover the early events in our B cell-driven model of autoimmunity, we performed immunofluorescence microscopy, demonstrating the presence of idiotype bright cells in the EFSBCs already at 4 weeks post reconstitution, in close juxtaposition with CD4 positive cells and DCIR2+ dendritic cells (Supplementary Fig. 3). This was well in line with a previous study reporting that extrafollicular B-cell activation by DCIR2+ marginal zone-associated CD8α- dendritic cells could drive T cell–dependent antibody responses[36]. In that study, antigen-specific B cells primed by DCIR2+ DCs were seen to accumulate within EFSBCs as early as 24 hours after immunization and were remarkably efficient at driving naive CD4 T-cell proliferation. However, these responses only formed GCs and displayed affinity maturation if TLR7 or TLR9 agonists were included at the time of immunization. In the context of autoreactivity, this fits well with a previous suggestion that the response initially focuses on self-components for which B cells carry TLR ligands, exactly because these are uniquely able to activate B cells independently of T cells, with subsequent T–B interactions activating autoreactive T cells, resulting in chronic autoimmunity[48]. Based on our observations in the GC blocked 564Igi driver model (Fig. 2), we cannot, however, definitively say whether a potential B-cell driven instruction of the T-cell repertoire occurs in the EFSBCs, extrafollicular foci, or in the primary (pre-GC) foci.

Others have previously attempted to determine if induction of autoreactive B cells allows priming of autoreactive T cells, but this was performed in an artificial model antigen setting[49,50]. In a more physiologically relevant model of autoimmunity, the antigen-presenting capabilities of B cells were found to be required for severe and lasting disease[2], however, this and other prior studies could not control for the presentation of ubiquitous endogenous self-antigen by other subsets of antigen-presenting cells. Accordingly, we considered the possibilities for directly interrogating the critical interaction of initiator B cells with T cells in a complex autoreactive environment in vivo and arrived at a setup in which the 564Igi driver compartment (MHC[b/b]) and a compatible WT compartment (MHC[b/b]) was separated from a second WT compartment (MHC[d/d]) by an MHC barrier (Fig. 4). Both the human leukocyte antigen (HLA) locus, encoding human MHC variants, and the H2 locus of wild mice are polygenic, highly polymorphic, and MHC variants are co-dominantly expressed, creating a broad basis for diverse peptide presentation. Conversely, inbred murine models of autoimmune disease are commonly monomorphic for MHC, restricting the breadth of the T-cell repertoire engaged in the autoimmune process. To evaluate the break-of-tolerance and subsequent epitope spreading in a more physiological setting, we generated mixed chimeras carrying multiple MHC haplotypes in an autoreactive milieu dominated by a diverse WT repertoire of B and T-cell receptors. In this complex and physiologically relevant autoreactive environment

in vivo, we demonstrate that B cells, by virtue of their capacity as antigen-presenting cells, are critically involved in establishing and driving the spread of the autoimmune response (Figs. 4–7).

We demonstrate a cellular relay mechanism, whereby the auto-reactive driver cells can break tolerance in a syngeneic MHC-restricted T-cell compartment, and in turn among other syngeneic B-cell clones. However, they cannot break tolerance in an allogenic T and B-cell compartment, unless their interaction is bridged by a B-cell compartment harboring both MHC haplotypes (Figs. 5, 6). Here, we emphasize specifically that it is the B-cell compartment, because it is the only possibility that fully explains our observations. In the three-way no bridge BM chimeras, we have myeloid cells representing both of the allotypes present. If their uptake of immune complexes and subsequent antigen presentation in the inflammatory environment were sufficient to directly break tolerance, we would expect that there was no barrier to the spreading of autoimmunity. This is because dendritic cells of the MHC[b] and MHC[d] haplotype experience the same environment and are exposed to the same immune complexes, and hence would be equally able to take up autoantigens and present these to T cells in an activating context. The B cells on the other hand, are unique in requiring BCR engagement for antigen uptake–i.e., the uptake and presentation of antigen to T cells is antigen-specific. This means that a scenario, where half the B cells express one MHC allotype and half another, is fundamentally different from a scenario, where all the B cells express both. Accordingly, if the initial activation originated within a B cell, and this B cell communicated with T cells, this would in turn be able to communicate with half the population in the former scenario, but all of it in the latter. Of note, prior work elegantly employed transgenic BCR mice lacking secreted antibody to isolate B-cell antigen presentation in the absence of Fc-receptor mediated uptake of soluble immune complexes by other antigen-presenting cells[51]. However, Fc-receptors are not the only critical aspect of antigen uptake and transport. This can be accomplished by complement receptors as well[52], a factor which could not be ruled out in that study, because the B-cell receptor itself can efficiently activate complement[53], and activation can additionally occur antibody independently[54].

Although the driver clone may act T-cell independently, at least initially, it is crucial that T-cell tolerance has to be broken to elicit the observed GC responses. Our data demonstrate that the epitope spreading process occurs in GCs, where the inclusion of B cells of a particular MHC haplotype hinges upon the initial focus of the auto-antigenic response (Fig. 4G, I, J), unless this response has been allowed to expand through B cells carrying additional MHC haplotypes (Fig. 5G, I, J). Our model overcomes previously noted limitations of transgenic models of autoreactive B cells[55], as the epitope spreading process studied occurs in a complex pool of physiologically normal B cells. Our findings may help explain the efficacy of B-cell ablation in treating diverse autoimmune diseases such as multiple sclerosis and Type I diabetes (before fully evolved disease development). Furthermore, our findings suggest that narrow antigen-specific targeting of T-cell auto-reactivity will fail, and at the same time emphasize the importance of moving towards antigen-specific targeting of B cells to curb auto-immunity. Antigen-specific targeting of B cells is currently being explored, e.g., by using autoantigen drug conjugates, chimeric auto-antibody or autoantigen receptor (CAAR) T cells, or immunomodula-tory nanoparticles, to either downmodulate their activity via co-engagement of inhibitory receptors such as CD22, or to directly ablate these cells[56]. This is of relevance not only in a plethora of intractable autoimmune conditions, but also in the context of checkpoint inhi-bitor therapy, where autoimmune related adverse events are the main cause of treatment cessation. Conversely, it also opens an entirely novel avenue of possibilities, such as harnessing the antigen-presenting capabilities of B cells to focus immune responses on tumor neo-antigens and thereby guiding T-cell directed responses against these.

# Methods

## Ethics statement

All animal experiments were conducted in accordance with the guidelines of the European Community and were approved by the Danish Animal Experiments Inspectorate (protocol numbers 2017-15-0201-01348 and 2017-15-0201-01319).

## Experimental model and subject details

CD45.1 congenic mice (B6.SJL-*Ptprc*$^a$ *Pepc*$^b$/BoyJ), CD45.1 congenic PA-GFP reporters (B6.Cg-*Ptprc*$^a$ Tg(UBC-PA-GFP)1Mnz/J)[57], H2d congenic mice (B6.C-*H2*$^d$/bByJ), and mice deficient in all MHCII genes, MHCII$^{-/-}$ (B6.129 S2-*H2*$^{dlAb1-Ea}$/J)[58] were obtained from Jackson Laboratories. C57BL/6JRj mice were obtained from Janvier Labs. The 564Igi line (B6.129S4(Cg)-*Igk*$^{tm1(Igks64)Tik}$ *Igh*$^{tm1(Ighs64)Tik}$/J)[29] was kindly made available by Thereza Imanishi-Kari, Tufts University, and provided by Michael C. Carroll, Boston Children's Hospital. Activation-induced cytidine deaminase (Aicda)-Cre transgenic mice on a C57BL/6 background were kindly provided by Meinrad Busslinger, Research Institute of Molecular Pathology, Vienna Biocenter, and are described in[59]. B6.129(Cg)-FoxP3$^{tm4(YFP/icre)Ayr}$/J[60] and B6.129S(FVB)-Bcl6$^{tm1.1Dent}$/J[61] were bought from Jackson Laboratories (JAX stock #016959 and #023727, respectively). Both Aicda-Cre and FoxP3-cre lines were crossed separately in-house with the Bcl6$^{flx/flx}$, and Aicda-Cre Bcl6$^{flx/flx}$ were subsequently crossed to 564Igi, to generate the experimental mice. Mice were housed in the Animal Facility at the Department of Biomedicine, Aarhus University, Denmark, under specific pathogen-free (SPF) conditions in individually ventilated cages at ambient temperature of 20–22 °C and ambient humidity, on a 12-h light/dark cycle with standard chow and water *ad libitum*. Both male and female mice were used in experiments. BM recipients were 8-18 weeks old at irradiation and donors were 7–22 weeks old at the time of BM harvest.

## Bone marrow chimeras

Recipient mice were irradiated either with 11 Gy gamma-irradiation (Gamma-cell 2000 RH, Model AK, Risø, with dose-time calculated according to the latest dosimetry measurements and correction according to decay table) or with 9 Gy (internal dosimetry) in a MultiRad 350 (Faxitron), with 350 kV, 11.4 mA, a Thoraeus filter [0.75 mm Tin (Sn), 0.25 mm Copper (Cu), and 1.5 mm Aluminum (Al)], and with a beam-distance of 37 cm. These methods have previously been verified to yield comparable results in connection with a nation-wide transition from nuclear to non-nuclear sources in Denmark[27,62]. Irradiated recipients were kept on antibiotic water (either 1 mg sulfadiazine together with 0.2 mg trimethoprim per mL drinking water, or 0.25 mg amoxicillin per mL drinking water) to avoid opportunistic infections. On the following day, donor mice were anesthetized with continuous flow of 3% isoflurane and euthanized by cervical dislocation. Femora, fibulae/tibiae, ossa coxae and humeri were harvested, mechanically cleaned, and rinsed in FC buffer (PBS, containing 2% heat-inactivated fetal calf serum (FCS), and 1 mM ethylenediaminetetraacetic acid (EDTA)). The bones were crushed in a mortar to release the bone marrow (BM) cells, and the cell extract was then passed through a 70 µm cell strainer. The donor BM cells were counted in a Cellometer K2 cell counter (Nexcelom) using Acridine orange (AO) and propidium iodide (PI) (ViaStain AOPI staining solution). Donor marrow was depleted of NK and T cells using biotinylated anti-NK1.1, TCRβ, CD3ε, CD4, and CD8a, followed by magnetic separation using streptavidin microbeads according to the manufacturer's instructions (MojoSort, BioLegend). Cells from the desired combinations of mice were then mixed according to the proportions mentioned in the figure legends, pelleted by centrifugation (200 $g$, 10 min, 4 °C) and resuspended to $1 \times 10^8$ cells/mL. The donor cell mixtures were used to reconstitute the recipient mice by retroorbital injection of 200 µL (containing a total of 20*10$^6$ cells) into each recipient mouse. Cohorts were analyzed at the times post reconstitution indicated in the figure legends.

## Treatment with clone 564 C11, 9D11 or mIgG

The hybridomas producing clone 564 C11 (Idiotype, Id) and clone 564 9D11 (anti-idiotype, αId), were kindly provided by Elisabeth Alicot, Boston Children's Hospital, and grown in-house. Mice were injected with 100 µg C11, 9D11, or purified mIgG per mouse i.p. two times/week for 5 weeks starting at day zero. When in combination with irradiation and BM reconstitutions, as described above, the C11 treatment was initiated on the same day as reconstitution.

## Tissue harvest and preparation

Following anesthesia in continuous flow of 3% isoflurane, blood was collected from the retroorbital plexus via a microcapillary tube into Eppendorf tubes containing 200 µl PBS with 5 mM EDTA. At end-point, mice were anesthetized and either bled retroorbitally followed by cervical dislocation, or mice were decapitated followed by blood collection. Spleen, IngLN, and MesLN were harvested, and the spleen was divided using surgical scissors. Tissues were either placed into ice-cold FC buffer for flow cytometry analysis or prepared for histology. For flow cytometry, the spleen and lymph nodes in ice-cold FC buffer were mechanically dissociated using pestles. Samples were filtered through 70 µm cell strainers. Spleen samples were centrifuged at 200 $g$ for 5 minutes at 4 °C, lysed in RBC lysis buffer (155 mM NH$_4$Cl, 12 mM NaHCO$_3$, 0.1 mM EDTA), incubated at RT for 3 minutes, centrifuged, and finally resuspended in FC buffer. For histology, spleen slices were placed directly into cryo-molds containing OCT (4583, Sakura Finetek) and fresh-frozen.

## Flow cytometry

Twenty µL Fc-block (553142, BD) diluted 1:50 in FC buffer and 100 µL of each sample was added onto a 96-well plate and then incubated for 5–10 min on ice. Antibodies (Supplementary Table 1) and fixable viability dye (65-0865-14, ThermoFisher Scientific) were diluted 1/300–1/500 and 1/2000, respectively, in FC buffer. One hundred µL antibody mix was added to each sample well and incubated for 30 minutes on ice. The plate was centrifuged at 200 $g$ for 5 min, the supernatant was removed, and cells were fixed for 30 minutes in PBS, 0.9% formaldehyde (F1635, Sigma-Aldrich) at RT. Following fixation, the plates were centrifuged at 200 $g$ for 5 min, the supernatant discarded, and the samples resuspended in FC buffer. Flow cytometry evaluation was performed the following day using a 4-laser (405 nm, 488 nm, 561 nm, 640 nm) LSRFortessa analyzer (BD instruments) or Novocyte Quanteon. Data were analyzed in FlowJo version 10.8.1 with UMAP v. 3.1[63], or FCS Express v. 7. The various gating strategies are summarized in Supplementary Figs. 9-12 and in Supplementary Table 2.

Fractional GC participation for MHCII experiments in Supplementary Fig. 2 was calculated as:

$$\frac{CD45.2[GCB]}{CD45.2[B]}*100\% \tag{1}$$

Fractional GC participation for MHC haplotype experiments in Figs. 4 and 5 was calculated as:

$$\left( \frac{I-A^d[GCB]}{I-A^d[GCB]+I-A^b[GCB]} \right) \Big/ \left( \frac{I-A^d[B]}{I-A^d[B]+I-A^b[B]} \right)*100\% \tag{2}$$

Normalization of MFI for I-A$^b$ and I-A$^d$ signal, respectively, within GCB or B-cell compartments of each tissue was performed according to the following formulae:

$$\frac{MFI \; of \; I-A^b \left[ I-A^{x/x} \right]}{MFI \; of \; I-A^b \left[ I-A^{b/b} \right]} \tag{3}$$

And

$$\frac{MFI\ of\ I - A^d \left[ I - A^{x/x} \right]}{MFI\ of\ I - A^d \left[ I - A^{d/d} \right]} \qquad (4)$$

## Immunofluorescence analyses of spleen sections

A Cryostar NX70 Cryostat (Fisher Scientific) was used to cut 16 or 20 μm thick spleen sections which were mounted on SuperFrost+ glass slides (ThermoFisher Scientific). Spleen sections were either acetone or PFA fixed. For acetone fixation, the spleen sections were fixed in acetone for 10 minutes at room temperature (RT), whereafter the slides were rehydrated in PBS for 3 minutes. For PFA fixation, the slides were rinsed with PBS, fixed with 4% w/v PFA for 30 min at RT, incubated in TBS (10 mM Tris, 140 mM NaCl, pH 7.4) for 30 min at RT, rinsed briefly with PBS, and incubated with permeabilization buffer (PBS, containing 2% v/v FBS, 0.1% w/v sodium azide, 0.1% v/v Triton-X100) for 45 min at RT. Antibodies were diluted in staining buffer (PBS, 2% v/v FBS, 0.1% w/v sodium azide). The antibody mix was centrifuged at $10,000\,g$ for 5 min and added onto the spleen samples, where it incubated overnight at 4 °C. The slides were washed once with staining buffer for 5 min and washed thrice in PBS with 0.01% v/v Tween-20 for 5 min. Slides were spot-dried and mounted using Fluorescence Mounting Medium (S3023, Dako).

Sections in Supplementary Fig. 3 were acetone fixed and all antibodies were diluted 1:500: CD8α-biotin, Streptavidin-BV421, CD138-PE, B220-AlexaFluor647, 33D1-AlexaFluor488 (A, B), CD4-A647, CD11c-BV421, 33D1-AlexaFluor488, CD169-PE (C-E), 9D11-AlexaFluor647, IgD-PacificBlue, 33D1-AlexaFluor488, CD169-PE (F), B220-Pacific Blue, 9D11-AlexaFluor647, 33D1-AlexaFluor488, CD138-PE (G), CD4-Qdot605, 9D11-AlexaFluor647, B220+biotin, Streptavidin-QD705, CD169-PE, 33D1-AlexaFluor488, CD11c-BV421 (H-J), CD4-Qdot605, 9D11-AlexaFluor647, IgD-AlexaFluor488, I-Ab-BV421, CD11c+biotin, Streptavidin-Qdot705 (K-N), 9D11+biotin, Streptavidin-Qdot705, CD4-Qdot605, CD138-PE, 33D1-AlexaFluor488, CD169-AlexaFluor647, and B220-pacific blue (O).

For Fig. 2, IgD-Pacific blue (Q-S; 1:300 and T-W: 1:250), Ki67-eFlour 660 (1:500), CD45.2-PE (1:500), CD3ε-Biotin (1:500), Streptavidin-FITC (1:500), CD138-PE (1:300), CD169-AF488 and 9D11-iFlour647 (1:500) were used as indicated in the panels. For quantification of GC formation in Figs. 4 and 5, sections were PFA fixed and the following antibodies were used: Ki67-eflour660 clone SolA15 (50-5698-82, ThermoFisher Scientific, 1:500), IgD-PacBlue clone 11-26c-2a (dilution 1:300), I-Ab-PE clone AF6-120.1 (dilution 1:500), and I-Ad-FITC clone 39-10-8 (dilution 1:300).

Imaging was performed using either: an Olympus VS120 Upright Widefield fluorescence slide scanner equipped with a digital monochrome camera (Hamatsu ORCA Flash4.0V2) and a 2/3" CCD camera, as well as single-band exciters and a filter wheel with single-band emitters (Hoechst, FITC, Cy3, Cy5, and Cy7); or a Zeiss LSM800 4-laser (405, 488, 561 and 640 nm) confocal microscope fitted with a 40x NA 0.95 air Plan Apo objective and equipped with Airyscan. Olympus OlyVIA 3.12 and Fiji v. 2.1.0/1.53c[64] were used for image processing. Channel intensities were uniformly adjusted for visual clarity in represented micrographs, but quantification was performed on raw images throughout.

## Anti-dsDNA measurements by time-resolved immunofluorometric assay (TRIFMA)

A FluoroNunc Maxisorp 96-well plate was coated with 100 μg/mL salmon sperm dsDNA (AM9680, ThermoFisher Scientific) in PBS and incubated overnight at 4 °C. Wells were blocked with 200 μL TBS containing 1% bovine serum albumin (BSA) (A4503, Sigma-Aldrich) for 1 hour at RT and washed thrice with TBS/Tw (TBS containing 0.05% v/v

Tween-20 (8.17072.1000, Merck)). Samples, standards, and quality controls were diluted in TBS/Tw containing 5 mM EDTA and 0.1% w/v BSA, and subsequently added in duplicates. The plate incubated at 37 °C for 1 hour. Wells were washed thrice in TBS/Tw and incubated with biotinylated goat-anti-mouse Ig (1010-08, Southern Biotech), 1 μg/ml TBS/Tw, at 37 °C for 1 hour. Wells were washed 3 times in TBS/Tw, and Eu³⁺-labeled streptavidin (1244-360, PerkinElmer) diluted 1:1,000 in TBS/Tw containing 25 μM EDTA was subsequently added to the wells and incubated at RT for 1 h. Finally, the wells were washed thrice with TBS/Tw, and 200 μL enhancement buffer (AMPQ99800, Amplicon) was added. The plate was shaken for 5 min and counts were read in a time-resolved fluorometry plate reader (Victor X5, PerkinElmer).

## Measurement of 564 C11 (Id) by TRIFMA

A FluoroNunc Maxisorp 96-well plate was coated with 1 μg/mL 9D11 in PBS and incubated overnight at 4 °C. Wells were blocked with 200 μL TBS containing 1 mg/ml I (Human Albumin CSL 109697) for 1 hour at RT and washed thrice with TBS/Tw (TBS containing 0.05% v/v Tween-20 (8.17072.1000, Merck)). Samples and standard (purified C11 were diluted in TBS/Tw containing heat agg. nhIg 100 μg/ml (Octapharma 478393), incubated 63 °C 30 min., and subsequently added in duplicates. The plate was incubated at 4 °C overnight. Wells were washed thrice in TBS/Tw and incubated with biotinylated goat-anti-mouse IgG2b (1093-08, Southern Biotech), or biotinylated goat-anti-mouse IgG2c (1077-08, Southern Biotech) or biotinylated mouse-anti-mouse IgM(b) (553519 BD), 1 μg/ml TBS/Tw, at RT for 2 hours. Wells were washed 3 times in TBS/Tw, and Eu3 + -labeled streptavidin (1244-360, PerkinElmer) diluted 1:1,000 in TBS/Tw containing 25 μM EDTA was subsequently added to the wells and incubated at RT for 1 hour. Finally, the wells were washed thrice with TBS/Tw, and 200 μL enhancement buffer (AMPQ99800, Amplicon) was added. The plate was shaken for 5 minutes and counts were read in a time-resolved fluorometry plate reader (Victor X5, PerkinElmer).

## Measurement of 564 9D11 (anti-Id) by TRIFMA

A FluoroNunc Maxisorp 96-well plate was coated with 1 μg/mL C11 in PBS and incubated overnight at 4 °C. Wells were blocked with 200 μL TBS containing 1 mg/ml I (Human Albumin CSL 109697) for 1 hour at RT and washed thrice with TBS/Tw (TBS containing 0.05% v/v Tween-20 (8.17072.1000, Merck)). Samples and standard (purified 9D11 were diluted in TBS/Tw containing heat agg. nhIg 100 μg/ml (Octapharma 478393, incubated 63 °C 30 min), and subsequently added in duplicates. The plate was incubated at 4 °C overnight. Wells were washed thrice in TBS/Tw and incubated with biotinylated goat-anti-mouse IgG1 (1073-08, Southern Biotech), 1 μg/ml TBS/Tw, at RT for 2 hours. Wells were washed 3 times in TBS/Tw, and Eu³⁺-labeled streptavidin (1244-360, PerkinElmer) diluted 1:1,000 in TBS/Tw containing 25 μM EDTA was subsequently added to the wells and incubated at RT for 1 hour. Finally, the wells were washed thrice with TBS/Tw, and 200 μL enhancement buffer (AMPQ99800, Amplicon) was added. The plate was shaken for 5 minutes and counts were read in a time-resolved fluorometry plate reader (Victor X5, PerkinElmer).

## Digital droplet PCR

B cells were purified from spleens of MHCII^b/b, MHCII^b/null, and MHCII^null/null mice using magnetic separation with a B cell isolation kit according to the manufacturer's instructions (130-090-862, Miltenyi Biotec). RNA was purified using TRI Reagent and the Direct-zol RNA Microprep kit (R2061, Zymo Research), and cDNA was generated using iScript cDNA synthesis kit (1708890, Bio-Rad). Digital droplet PCR was performed on a QX200 (Bio-Rad), multiplexing a FAM GAPDH assay (dMmuCPE5195282, Bio-Rad) and H2-Aa primer and HEX probe set (forward: CAATTGGCAAGCTTTGACCCC, reverse: TTGGGGAACACAGTCGCTTG, probe: [HEX]CCACCCCAGCTACCAAT GAGGC[MGBEQ]).

## Single-cell mRNA and repertoire sequencing

For single-cell mRNA and repertoire sequencing, total splenocytes were purified from spleens of two no bridge chimeras as indicated under Tissue Harvest and Preparation, followed by negative selection with biotinylated Ter-119, anti-biotin microbeads and magnetic separation. For one bridge chimera, the spleen was split in two, and one third was subjected to the same total splenocyte purification, whereas two thirds of the spleen was used for purification of GC B cells using a PNA microbead kit (130-110-479, Miltenyi Biotec) according to the manufacturer's instructions. The two resulting fractions were mixed to generate a GC B cell-enriched splenocyte sample. From each sample, $1 \times 10^6$ cells were added 100 μl of AbSeq staining mix containing a cocktail of 20 Abseq antibodies and incubated for 60 minutes. Cells were washed three times with 2 ml BD staining buffer (250 $g$ for 8 minutes centrifugation) and for each sample, 20,000 cells were loaded onto a BD Rhapsody cartridge and processed on the Rhapsody Express module following the manufacturer's instructions. Each sample was subsequently split into two (A and B) to be able to verify technical reproducibility, and each subsample was used for generation of 4 libraries: (1) single-cell targeted library using the Mouse Immune Response Panel (Catalog No. 633753, BD Biosciences) containing primer pairs targeting 397 genes commonly found in mouse immune cells and a BD Rhapsody Panel supplement (Part number 633742, BD Biosciences) containing an additional 45 target genes, for a total of 442 targets; (2) Abseq library; (3) BCR V(D)J library; and (4) TCR V(D)J library. BCR and TCR V(D)J primers were as given in the BD Mouse VDJ CDR Library Preparation Protocol (https://scomix.bd.com/hc/en-us/articles/360039007471-VDJ-CDR3-Protocols, Doc ID: 23-22687-00, Revision 05/2020). TCR:BCR:targeted panel libraries were pooled at a pooling ratio of 40:50:10 and sequenced on a NovaSeq 6000 with SP, 300, using 15% PhiX spike-in. Abseq libraries were sequenced using PE100 sequencing on DNBSEQ-G400.

## Single-cell mRNA and repertoire data analysis

The raw sequencing data were processed on the Seven Bridges Genomics platform using the BD Rhapsody Targeted Analysis Pipeline and analyzed in R v. 4.1.2[65]. To further ensure the quality of the dataset, the cells with more than 1000 RNA count (doublets) and the cells with less than 25 detected genes (debris) in gene expression matrix were removed. Afterwards the data were further cleaned up with Doublet-Finder v. 2.0.3 to remove doublets[66]. After filtering, cells from different libraries were merged and unsupervised clustering was performed using Seurat v. 4.0.6[67]. Briefly, variable genes were determined using Seurat's FindVariableGenes function with default parameters (selection.method = "vst", nfeatures = 2000). Clusters were identified via the FindClusters function (resolution 0.2) in Seurat using principal components and subsequently visualized using the RunTSNE function (reduction = "pca"). tSNE and UMAP plots were applied for visualization. After annotation, T cells and B cells were subset individually and repertoire data for each single cells were assigned to each cell. Only the cells with repertoire information were filtered and used for downstream subcluster analysis. Cells were gated with SeqGeq 1.7.0 (BD Biosciences) based on Abseq expression.

## Bead-based antigen arrays

A custom bead-based antigen array was created[68]. The panel included 71 antigens, including cytokines, chemokines, growth factors, acute phase proteins, cell surface proteins, antigens associated with connective tissue disease, and antigens from various viral pathogens. Each antigen was coupled to carboxylated magnetic beads (MagPlex-C, Luminex Corp.) with a unique barcode. For each assay, the bead array was distributed into a 384-well plate (Greiner BioOne) by transferring 5 μl of bead array per well. 45 μl of the 1:100 diluted sera in 0.05% PBS-Tween supplemented with 1% w/v bovine serum albumin were transferred into the 384-well plate containing the bead array. Samples were

incubated for 60 min on a shaker at room temperature. Beads were washed 3 times with 60 μl PBS-Tween on a plate washer (EL406, Biotek), and 50 μl of 1:1000 diluted R-phycoerythrin (R-PE) conjugated Fc-γ-specific goat anti-mouse IgG F(ab')2 fragment (Jackson ImmunoResearch) was added to the 384-well plate. After a 30-minute incubation, the plate was washed with 3 times with 60 μl PBS-Tween and resuspended in 50 μl PBS-Tween prior to analysis using a FlexMap3D™ instrument (Luminex Corp.). Binding events were displayed as Median Fluorescence Intensity (MFI). For normalization, MFI values for "bare bead" IDs were subtracted from MFI values for each antigen-conjugated bead ID for each sample. Samples were run in duplicate.

## Autoantigen pull-downs

Autoantigen pull-downs were based upon a prior protocol[44]. At 6-8 weeks of age, four female C57BL/6J mice were euthanized by cervical dislocation. The spleens were extracted, and each placed into 600 μl buffer consisting of 20 mM imidazole, 10% w/v sucrose, 2 mM EDTA, pH 7.4, containing 1× protease inhibitor cocktail (Thermo Fisher 78425), on ice. Spleens were then homogenized in a bullet blender (Next Advance) at speed 8 for 30 seconds, then centrifuged at 3,000 $g$ for 10 min, at 4 °C. The supernatant was aliquoted and frozen at −20 °C until use. For pre-clearing of supernatants, sepharose beads (Cytiva 17088601 Gammabind Sepharose) were washed in Buffer 3 (0.2 M glycine, pH 2.5), then equilibrated in Buffer 1 (PBS containing 0.05% Tween-20, 0.1% w/v BSA, 0.1% NaN$_3$ and 2 mM EDTA). Subsequently, 350 μl beads were incubated with 350 μl pooled spleen supernatant and 350 μl Buffer 1, for 30 min, end-over-end at 4 °C. Beads were pelleted at 100 $g$ for 3 min. at 4 °C, and the supernatant extracted. The pre-cleared supernatant was then aliquoted into 50 μl aliquots, which each received either 100 μg purified IgG (clone 564 C11, clone 564 9D11, or murine serum IgG) or 10 μl of serum (a pool of serum from 6 C57Bl/6 J mice, a pool of serum from 5 564Igi homozygous mice, or individual sera from no-bridge or bridge chimeras). The mixtures were incubated 1 h on ice, then each was added 25 μl washed and equilibrated Sepharose beads and incubated another 30 min end-over-end at 4 °C. Beads were recovered by centrifugation at 100 $g$, for 3 min at 4 °C, then washed 4 times with 500 μl Buffer 2 (PBS containing 0.05% v/v Tween-20, 0.1% NaN$_3$, and 2 mM EDTA). Beads were again recovered by centrifugation, then eluted with 55 μl Buffer 3, and the eluate neutralized with 13 μl 1 M Tris-HCl, pH 8.0. Eluates were frozen at −20 °C until further analysis.

## LC-MS analyses of proteins

The label-free proteomics analyses were performed similarly to a previous protocol[69]. Briefly, 25 μl eluate was mixed with SDS-PAGE sample buffer, and subjected to SDS-PAGE followed by reduction, blocking of reduced cysteine residues, and in-gel trypsin digestion. The resulting peptides were purified with PepClean C18 Spin Columns and vacuum dried. Peptide samples were trapped and separated by liquid chromatography (Easy-nLC 1200, Thermo Scientific) using pre-column (Acclaim PepMap 100, 75 μm × 2 cm, Nanoviper, Thermo Scientific) and analytical column (EASY-Spray column, PepMap RSLC C18, 2 μm, 100 Å, 75 μm × 25 cm) in a 60 min gradient of 4–40% acetonitrile in 0.1% formic acid, coupled to the mass spectrometer (MS) Q-Exactive HF-X Hybrid Quadrupole Orbitrap (Thermo Scientific, Bremen). Full scan (MS1) resolution was set at 60,000 (at 200 m/z) and the scan range was between 372 and 1800 m/z. Up to the 12 most intense peaks in MS1 were fragmented in MS2 using data dependent acquisition. MS2 resolution was set at 15,000. Unassigned and +1 charge state was excluded from fragmentation and a dynamic exclusion of 15 s was used. Proteins were identified and quantified using MaxQuant[70] (version 1.5.3.30) with its Andromeda algorithm against a mouse sequence database (*Mus musculus* proteome with 17,544 reviewed sequences downloaded from Uniprot database, https://www.uniprot.org/ on April 12th 2022). The "match between runs" was applied and iBAQ data, were used for quantification. Data were filtered to exclude IgH, IgK,

and IgL chains derived from the pull-down antibodies. Detected keratin proteins were found to stem from contamination from human and were therefore excluded from the analyses. For purified immunoglobulin pull-downs, all iBAQ values were added 1 then log2 transformed before volcano plotting. For serum immunoglobulin pull-downs, raw iBAQ values were imported in R (v. 4.1.2), and then scaled using the scale function. One column with missing data (ATP-dependent RNA helicase A) was removed, principal components were calculated using the prcomp function of FactoMineR (v. 2.8), and a PC1 and PC2 plot was generated. A heatmap was generated using the native heatmap function in R.

### Statistical analyses

GraphPad Prism v. 9 was used for statistical analyses. The nature of tests, $n$, and the meaning of bars, lines, and error bars are indicated in the figure legends. Exact $p$-values are given in figures throughout, except when $p > 0.05$ (ns) or when $p < 0.0001$ ($p < 0.0001$, no exact $p$-value calculated).

### Reporting summary

Further information on research design is available in the Nature Portfolio Reporting Summary linked to this article.

## Data availability

The data that support this study are available from the corresponding author upon request. Processed data and raw data for single-cell RNA sequencing produced in this study are available via the Gene Expression Omnibus (GSE202358). We have furthermore established easily accessible databases for the total dataset: (https://dreamapp.biomed.au.dk/B_cells_inclusion_TOTAL/), and B cell dataset: (https://dreamapp.biomed.au.dk/B_cells_inclusion_B_cells/). The mass spectrometry proteomics data have been deposited to the ProteomeXchange Consortium via the PRIDE[71] partner repository with the dataset identifier PXD044323). Raw data for the autoantigen array screenings have been provided with Source Data and additional raw data shown in the figures are provided in the Source Data file. Source data are provided in this paper.

## Code availability

No custom code has been used.

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

## Acknowledgements

S.E.D. was supported by the Novo Nordisk Foundation (NNF17OC0028160 and NNF19OC0058454), the Lundbeck Foundation (R238-2016-2954), the LEO Foundation (LF-OC-22-000977), and the Independent Research Fund Denmark through the DFF Danish ERC support programme (8124-00001B) and a Sapere Aude Research Leader grant (9060-00038B), as well as a Carlsberg Foundation Distinguished Fellowship (CF18-0446). S.E.D. and A.F. are members of and were supported by the Danish National Research Foundation Centre of Excellence, CellPAT, Center for Cellular Signal Patterns (DNRF135). C.F-H was supported by a postdoctoral fellowship from Lundbeckfonden (R303-2018-3415). Y.L. was supported by the European Union's Horizon 2020 research and innovation programme (899417) and the Novo Nordisk Foundation (NNF21OC0068988 and NNF21OC0071031). L.L. was

supported by an Independent Research Fund Denmark—Sapere Aude Starting Grant (8048-00072 A), and the Novo Nordisk Foundation (NNF21OC0071718). P.J.U. was supported by the National Institute of Allergy and Infectious Diseases of the National Institutes of Health (R01 AI125197-04), philanthropic support from the Sean N Parker Center COVID-19 Research Fund, and the Henry Gustav Floren Trust. The mass spectrometry analyses were supported by John and Birthe Meyer Foundation (J.P.). Flow cytometry was performed at the FACS Core Facility, Aarhus University, Denmark. The authors would like to thank Charlotte Christie Petersen, Anja Bille Bohn, and Anni Skovbo at the FACS Core AU for expert technical assistance with flow cytometry analyses. We would furthermore like to acknowledge AU Health Bioimaging Core Facility for the use of equipment and support of the imaging facility. We would like to thank Christian H. Garm for expert technical assistance with microscopy and image analysis.

## Author contributions

Conceptualization, S.E.D., C.F.-H., T.R.W.; Methodology, S.E.D., T.R.W., C.F.-H., E.T.-D., L.L, J.D.N., and J.P.; Investigation, S.E.D., T.R.W., C.F-H, E.T.-D., K.S.K., L.F.V, M.K.P, A.P., G.W., S.F., N.v.C., L.J., J.H., Y.L., L.L., E.Y., A.N.R., Q.R.M., C.E.vdP., and J.P.; Writing—original draft, S.E.D. and C.F.-H.; Writing—review and editing, T.R.W., E.T.-D., L.F.V., J.P., J.H., L.L.; Visualization. C.F.-H., S.E.D.; Supervision, P.J.U., M.C.C., L.L., and S.E.D.; Project Administration, T.R.W, C.F.-H. and S.E.D.; Funding acquisition: C.F.-H., S.E.D.

## Competing interests

The authors declare that no competing interests.

## Additional information

Ethics statementAll animal experiments were conducted in accordance with the guidelines of the European Community and were approved by the Danish Animal Experiments Inspectorate (protocol numbers 2017-15-0201-01348 and 2017-15-0201-01319).

