## [Peer Review File · Nature Communications]

REVIEWER COMMENTS

Reviewer #1 (expert in SLE, immune mechanisms of systemic autoimmune diseases):

In this paper Fahlquist-Hagert et al use the 564Igi mixed chimeric model to examine the immunologic mechanisms that lead to epitope spreading in a nuclear antigen-restricted immune response. The paper builds upon previous work by members of the group, where they showed that mice with knockin Ig heavy and light chains derived from a mutated high affinity Ig receptor with specificity for ssDNA and Ro60, when in a polyclonal repertoire (heterozygous Ig knockin mice or homozygous Ig knockin mixed chimeric mice), breach tolerance leading to recruitment of WT autoreactive B cells into germinal centers and the auto-antibody response. They have previously shown that this response is dependent upon T cells (through expression of CD40L). In this paper they have explored further the immunologic events that lead to the initial breach of tolerance and that result in epitope spreading. They provide data indicating that recruitment of 564Igi B cells into germinal centers is not required for the initial breach of tolerance and suggest that this occurs in the bridging channels between the red and white pulp. They then explore the mechanisms that lead to the breach of tolerance in these mice, providing several pieces of evidence to support the concept that this results from B cell presentation of autoantigens to T cells. Overall the paper is well-written and the data supports the conclusions made by the authors. The concept that T-B collaboration leads to epitope spreading is not particularly new, but the authors provide convincing evidence that in their model this clearly predominates over T-myeloid interactions. Several specific comments are outlined below:

1) Figure 2: In panel M the authors should show not only total IgG anti-dsDNA abs but also the results for IgG2a and IgG2c so that we can see how the autoantibody levels produced by 564Igi and WT cells are affected by the lack of recruitment of 564Igi cells into GCs. Also the organization of the micrographs in this figure is confusing. The micrographs should be re-ordered so that the stains are the same across each horizontal row (split channel on the left, Cre- and then Cre+). The figure legend should also be revised to better explain what panel R is meant to show, with a similar micrograph at the same magnification for Cre+.

2) The authors cite the work of Chappel et al (Ref 35) showing that extra-follicular B cells can be activated by DCIR2-positive marginal zone-associated CD8alpha negative DCs to initiate a T cell-dependent antibody response, and show data in Supplementary Figure 2 demonstrating co-localization of these DCs with CD138+564Igi+ cells and CD4+ T cells at 4 weeks post-reconstitution, suggesting that a similar mechanism could lead to activation of the B and T cells in their model. However, the data provided raises many unanswered questions. For example, how is tolerance in the 564Igi B cells overcome? Does this occur at a particular time point following reconstitution or occur throughout the life span of the mouse. In this context, some further phenotyping of these cells, including cell surface IgM, CD86, and expression of other activation/differentiation markers, would have been helpful. Additionally, is there evidence that Tfh or Tph cells become expanded at a specific time point following reconstitution? Previous work suggests that the extra-follicular B cell activation in their proposed mechanism is TLR independent, whereas the recruitment into GC following this kind of activation is TLR-dependent. Does this extra-follicular activation occur in the absence of TLR7?

3) In Supplementary Figure 2, some of the immune populations are very difficult to see at the magnifications shown, it would be helpful if some of the key findings were shown at higher resolution.

4) In Supplementary Figure 6, the differences in panels L and M are not statistically significant. This should be acknowledged in the text on page 14.

Reviewer #2 (expert in B cells and GC responses):

This is an interesting study on the role of B cells in the development of autoimmunity and epitope

spreading. The group has earlier shown that adoptive transfer of an autoreactive B cell clone leads to germinal centres and epitope spreading with the activation of other B cells leading to SLE like autoimmunity. In the current study they investigate the mechanisms behind this, showing that the initiating autoreactive clone itself is excluded from germinal centres, does not need to enter germinal centres, and also that antibody from the clone does not have a role in the activation of further autoimmune cells. What is important is antigen-presentation from the initiating clone. With a series of complex, but well-explained experiments the authors show that MHC-dependent information exchange travels from the initiating clone to T cells and other B cells. I find the study well done, and the results and conclusions convincing. The final in depth analysis of gene expression does not add very much to the conclusions, but this is not necessary. Conclusions are relevant for other autoimmune disease and may have implications on future work identifying targets for prevention or therapy.

Problems are that negative controls for Fig. 2 (showing that germinal centres form without 564Igi cells entering) and positive controls in Fig. 3 (showing that germinal centre do not form after antibody) are missing.

Further, throughout the study germinal centres, and sometimes plasma cells were used as a surrogate marker for autoimmunity. I would prefer to see evidence that GCs are surrogates for autoimmunity in the different experiments and conditions. Would it be possible to show this, e.g. by presence of autoreactive antibody?

Minor issues:

Some figures are very small and difficult to read when printed.

The graphical abstract is not clear for someone who is not expert or has not read the manuscript.

Can a legend be added? E.g. what are the pink star-shaped cells?

Fig. 1A legend: numbers under spleens: unit is missing

Line 147: "+" is usually used for wt. Would another symbol for knockin be better?

Line 148: K-/- means Ig kappa knockout?

Line 148 and Fig. 1B: I assume transfer was K+/- "or" K-/- B cells. Line 148 states "and".

Fig. 2A: Bcl: "flx" is missing

Suppl. Fig. 1G-I: I presume the GCs do not contain CD45.2 564Igi cells. Can this be shown?

Suppl. Fig. 3B legend: The time 6-8 wk and 12 wk do not fit with the single interval given in Fig. 4A, S3A, which is 8 wk. Why is that?

Reviewer #3 (expert in antigen presentation by B cells):

The authors present compelling evidence to prove that B cells initiate the autoimmune response outside of germinal center and epitope spreading occurs with antigen presentation dependent on MHC haplotype. The experiments are carefully designed to methodologically answer the scientific questions. Overall, the manuscript has valuable information that will promote the understanding of autoimmune diseases and the development of new treatment strategies. The article has some errors and require clarifications before acceptance for publication, as listed below:

1. It will be helpful to add a color-coded key to the graphs in figures 1, 2 and 4. Although it is evident that the colors are chosen according to the representation in the experiment schematic, adding the key for each graph will make interpretation easier for the readers.
2. In Figure 2, panel Q/R show the images from Cre- chimera, however the text in the result section states ".....verified the absence of CD45.2 B cells from GCs of Cre+ chimeras (Fig. 2O-R)". This needs to be rectified. [Also, I don't know if this was intentional, but there is no panel W in the figure].
3. In Figure 3, panel R has data showing Id antibodies in serum, however the text in the result section and the legend states "anti-dsDNA levels were comparable between the groups (Fig. 3Q and R)" and "Total IgG anti-dsDNA antibodies in sera of mice presented in panels M-Q" respectively. This needs to be rectified. [Also in the figure legend the panel "R" is mislabeled as

"M"]

4. In the discussion section the authors state that "Furthermore, our findings suggest that narrow antigen-specific targeting of T cell autoreactivity will fail, and at the same time emphasize the importance of moving towards antigen-specific targeting of B cells to curb autoimmunity". As this is one of the main take away messages from this manuscript the authors should discuss this in more detail i.e. by talking about current therapies that may or may not follow this approach and what could be the possible hurdles/shortcomings of this approach.

RESPONSE TO REVIEWERS' COMMENTS

First of all, we would like to thank the Reviewers for their thorough and careful review of our work. We were happy to see the overall positive reception, however, we also acknowledge that there were several comments that needed to be addressed. Accordingly, we have thoroughly revised the manuscript, and a brief overview of the main changes is given below:

- Prompted by comment of Reviewer 1, we have revised the nomenclature for 564Igi knock-in mice, replacing “+” with “ki” and “-” with “wt” throughout the manuscript and figures.
- Figure 1 was revised regarding nomenclature cf. the above, and ‘weeks’ post reconstitution was indicated in the schematic in panel A, as suggested by Reviewer 2.
- Figure 2 was revised to include additional isotype-specific TRIFMA data, switch order of imaging panels, and to add an additional panel for Cre+, as requested by Reviewer 1. We also added a clearer legend in panel A, as suggested by Reviewer 3, and corrected naming of 564Igi Aicda-Cre Bcl6 to include “flx/flx” in the schematic of panel A, as suggested by Reviewer 2.
- Figure 3: changed “FoxP3-cre Bcl6 mice” in schematic A to “FoxP3-Cre Bcl6flx/flx”, as prompted by Reviewer 2’s similar comment for Figure 2.
- Figure 4: added a clearer legend in panel A for the subsequent graphs, as suggested by Reviewer 3.
- NEW Figure 7 was added, presenting data from autoantibody pull-downs and mass-spec analyses.
- NEW summary Figure 8 was added, replacing graphical abstract, and updated with legend as requested by Reviewer 2.
- NEW Supplementary Figure 1 was added, to include additional controls for main Figure 2 and 3 as requested by Reviewer 2. Subsequent Supplementary Figures have been renumbered 2-7 and in-text references to these have been updated.
- Supplementary Figure 3 (previously 2) was revised to allow larger panel size as suggested by Reviewers 1 and 2.
- NEW Supplementary Figure 8 was added, to include additional data from the autoantibody pull-downs presented in new Figure 7.
- Two additional authors were added in connection with the revision work.
- Added descriptions of the additional controls in the results section.
- Highlighted central ongoing efforts towards antigen-specific targeting of autoreactive B cells in the discussion section and added a reference to a recent review on this topic, as suggested by Reviewer 3.
- Added description of the autoantibody pull-downs and MS-based analyses in the results section.
- Added additional subsections for the autoantibody pull-downs and MS-based analyses in the methods section.

Below, we provide a point-by-point response to the specific comments of the Reviewers.

REVIEWER COMMENTS

Reviewer #1 (expert in SLE, immune mechanisms of systemic autoimmune diseases):

In this paper Fahlquist-Hagert et al use the 564Igi mixed chimeric model to examine the immunologic mechanisms that lead to epitope spreading in a nuclear antigen-restricted immune response. The paper builds upon previous work by members of the group, where they showed that mice with knockin Ig heavy and light chains derived from a mutated high affinity Ig receptor with specificity for ssDNA and Ro60, when in a polyclonal repertoire (heterozygous Ig knockin mice or homozygous Ig knockin mixed chimeric mice), breach tolerance leading to recruitment of WT autoreactive B cells into germinal centers and the auto-antibody response. They have previously shown that this response is dependent upon T cells (through expression of CD40L). In this paper they have explored further the immunologic events that lead to the initial breach of tolerance and that result in epitope spreading. They provide data indicating that recruitment of 564Igi B cells into germinal centers is not required for the initial breach of tolerance and suggest that this occurs in the bridging channels between the red and white pulp. They then explore the mechanisms that lead to the breach of tolerance in these mice, providing several pieces of evidence to support the concept that this results from B cell presentation of autoantigens to T cells. Overall the paper is well-written and the data supports the conclusions made by the authors. The concept that T-B collaboration leads to epitope spreading is not particularly new, but the authors provide convincing evidence that in their model this clearly predominates over T-myeloid interactions. Several specific comments are outlined below:

We thank the Reviewer for the positive reception and the recognition that we provide convincing evidence for a long-standing concept of T-B collaboration as the basis for epitope spreading.

1) Figure 2: In panel M the authors should show not only total IgG anti-dsDNA abs but also the results for IgG2a and IgG2c so that we can see how the autoantibody levels produced by 564Igi and WT cells are affected by the lack of recruitment of 564Igi cells into GCs. Also the organization of the micrographs in this figure is confusing. The micrographs should be re-ordered so that the stains are the same across each horizontal row (split channel on the left, Cre- and then Cre+). The figure legend should also be revised to better explain what panel R is meant to show, with a similar micrograph at the same magnification for Cre+.

We have now measured anti-dsDNA IgG2a and IgG2c and included this in panel M of Figure 2. Again, the two bone marrow chimera groups did not differ significantly from each other. IgG2c levels were generally very low, trending slightly higher in the Cre- group. To further explore this, we additionally performed more sensitive anti-idiotypic antibody screenings on these sera, as a surrogate for autoreactive cells targeting ribonucleic acids (same idiotypic family). These data have been included in new Supplementary Figure 1, panels F-H. Idiotypic antibodies were slightly higher in Cre+ chimeras, a phenomenon which could be explained by the inability of 564Igi cells in Cre+ chimeras to partake in GCs and hence to hypermutate away from the idioype. Of note, the wild-type B cells entering autoreactive GCs likely target protein components of ribonucleic acid complexes to a higher degree than the nucleic acids themselves, as also suggested by our autoantigen profiling.

The micrographs in Figure 2 have additionally been reorganized as suggested by the Reviewer, and we have revised the explanation of panel R and included a similar micrograph for Cre⁺ (new panel S). Due to space constraints, we show only split-channel views for a single sample for each of the panels.

2) The authors cite the work of Chappel et al (Ref 35) showing that extra-follicular B cells can be activated by DCIR2-positive marginal zone-associated CD8alpha negative DCs to initiate a T cell-dependent antibody response, and show data in Supplementary Figure 2 demonstrating co-localization of these DCs with CD138+564Igi⁺ cells and CD4⁺ T cells at 4 weeks post-reconstitution, suggesting that a similar mechanism could lead to activation of the B and T cells in their model. However, the data provided raises many unanswered questions. For example, how is tolerance in the 564Igi B cells overcome? Does this occur at a particular time point following reconstitution or occur throughout the life span of the mouse. In this context, some further phenotyping of these cells, including cell surface IgM, CD86, and expression of other activation/differentiation markers, would have been helpful. Additionally, is there evidence that Tfh or Tph cells become expanded at a specific time point following reconstitution? Previous work suggests that the extra-follicular B cell activation in their proposed mechanism is TLR independent, whereas the recruitment into GC following this kind of activation is TLR-dependent. Does this extra-follicular activation occur in the absence of TLR7?

We agree with the Reviewer that the presented data opens new research avenues by raising numerous additional questions centered around the fundamental problem of how tolerance in the 564Igi B cells (or any autoreactive clone for that matter) is overcome. However, some of the sub-questions raised are partly answered by our previous work as outlined further below:

Does this occur at a particular time point following reconstitution or occur throughout the life span of the mouse?

Our previous work in the parental 564Igi strain showed that in the context of the chronic autoreactive GC B cell responses observed, a subset of GC B cells is very long-lived, up to at least 90 days (3 months) (see Aicda-CreERT2 LSL-YFP lineage tracing experiment in Figure 1 of (Degn et al., 2017)). This indicates that B cells that have broken tolerance at a very early time-point persist, however, at the same time demonstrates that some new cells ('dark', non-YFP) are entering the response. In the mixed chimera model, we observed the same phenomenon in our Confetti GC B cell lineage tracing experiments (Figure 5 of (Degn et al., 2017)), which was further corroborated by longitudinal observations of single GCs using an intravital lymph node imaging window (Firl et al., 2018). Importantly, the GC process proceeded, even if the initiating 564Igi clone was subsequently ablated (Figure 4 of (Degn et al., 2017)), showing that the response is co-opted by wild-type clones once T cell tolerance is broken. This is in line with previous observations by Hedda Wardemann and Michel Nussenzweig showing that many immature B cells are in fact autoreactive (Wardemann et al., 2003), and hence the 'potential for autoreactivity' is inherent in the emergent B cell population, but is limited by the environment under homeostatic conditions. This notion is further supported by recent work in pre-print, which demonstrates that naïve B cells frequently invade pre-existing spontaneous (autoreactive) GCs (van den Broek et al., 2023).

In this context, some further phenotyping of these cells, including cell surface IgM, CD86, and expression of other activation/differentiation markers, would have been helpful

We acknowledge the Reviewer's point that additional phenotyping would have been helpful, however, we were limited by the number of markers in our panel for flow cytometry. This could be solved by including an additional panel but would require a whole new series of experiments. We think this would be beyond the scope of the present, data-rich manuscript, which has already been expanded in this revision (now 8 main figures, 8 supplementary figures, and 2 supplementary tables). However, we would very much like to follow up the Reviewer's suggestion in future work.

Additionally, is there evidence that Tfh or Tph cells become expanded at a specific time point following reconstitution?

We have not performed kinetic analyses of Tfh or Tph specifically. Again, this would require an additional flow panel and a new set of chimeras. It is well known from work by Mark Ansel, Antonio Lanzavecchia and Federica Sallusto that antigen and GC B cells so to speak 'induce their own Tfh response' (Baumjohann et al., 2013). Indeed, as the Reviewer also notes, we previously characterized Tfh cells in the model and showed that ablation of Th cells by anti-CD40L antibody causes a collapse of the GC response (Figure S4 of (Degn et al., 2017)). Subsequent work has yielded an in-depth characterization of the Tfh landscape in the model (Akama-Garren et al., 2021). However, we are not confident that tracking a global increase in Tfh cells concomitant with GC emergence would shed light on the local effects we propose elicit the initial break-of-tolerance. More in-depth analysis of this would require extensive imaging (ideally longitudinal) and a series of functional experiments targeting the initial interaction between B cell and pre-Tfh cell. This is something we are actively working on, but which we feel is beyond the scope of the present manuscript.

Previous work suggests that the extra-follicular B cell activation in their proposed mechanism is TLR independent, whereas the recruitment into GC following this kind of activation is TLR-dependent. Does this extra-follicular activation occur in the absence of TLR7?

We do not have direct observations of the extra-follicular activation in a setting where B cells are deficient in TLR7. It is a major challenge that there is no specific marker (or marker combination) that uniquely identifies extra-follicularly activated B cells, making it impossible to positively identify these cells with absolute confidence. Based on our previous work in a setting where TLR7 deficient B cells compete head-to-head with TLR7 sufficient B cells, we observed an >90% reduction in TLR7 deficient GC B cells, but only ~75% reduction in TLR7 deficient plasma cell output (Green et al., 2021). Since the plasma cells are either derived from GC or extrafollicular responses, we may speculate that extrafollicular responses are less affected by the TLR7 block. The increased BCR signal afforded by the repeating antigenic structure of nucleic acid-associated targets may to some extent be able to compensate for the block in TLR7 signaling.

3) In Supplementary Figure 2, some of the immune populations are very difficult to see at the magnifications shown, it would be helpful if some of the key findings were shown at higher resolution.

We apologize for this. We have changed the layout of the figure to increase the panel size, within the limits of supplementary information in *Nature Communications*.

4) In Supplementary Figure 6, the differences in panels L and M are not statistically significant. This should be acknowledged in the text on page 14.

Our apologies for not having made this clear. Indeed, these differences did not reach statistical significance, likely due to the low n. We have now specified in the text on page 14 that the differences in Figure 6 panels L and M were not statistically significant, and we have modified the language to avoid overstating our claims. We thank the Reviewer for drawing our attention to this.

Reviewer #2 (expert in B cells and GC responses):

This is an interesting study on the role of B cells in the development of autoimmunity and epitope spreading. The group has earlier shown that adoptive transfer of an autoreactive B cell clone leads to germinal centres and epitope spreading with the activation of other B cells leading to SLE like autoimmunity. In the current study they investigate the mechanisms behind this, showing that the initiating autoreactive clone itself is excluded from germinal centres, does not need to enter germinal centres, and also that antibody from the clone does not have a role in the activation of further autoimmune cells. What is important is antigen-presentation from the initiating clone. With a series of complex, but well-explained experiments the authors show that MHC-dependent information exchange travels from the initiating clone to T cells and other B cells. I find the study well done, and the results and conclusions convincing. The final in depth analysis of gene expression does not add very much to the conclusions, but this is not necessary. Conclusions are relevant for other autoimmune disease and may have implications on future work identifying targets for prevention or therapy.

We thank the Reviewer for the positive reception of our work.

Problems are that negative controls for Fig. 2 (showing that germinal centres form without 564Igi cells entering) and positive controls in Fig. 3 (showing that germinal centre do not form after antibody) are missing.

We believe that we are presenting the most appropriate comparisons in the figures, but we see the Reviewer's point.

Specifically for Figure 2, the figure compares Cre⁺ and Cre⁻ driver chimeras side by side. We understand that by **negative controls** for the setup presented in Figure 2, the Reviewer refers to controls that do not present with germinal centers.

We had actually included C57Bl/6J controls and 564Igi homozygous Aicda-Cre⁺ vs. Cre- Bcl6flx/flx mice in the flow cytometry experiments presented in Figure 2, but these were intended to be the topic of a separate report and for that reason had not been included in the figure. We have now added these data in new Supplementary Figure 1 (panels A-E).

As can be seen, C57Bl/6J controls display no GC B cells in the IngLN and spleen, whereas ambient GCs are present in the MesLN (Fig. S1B). 564Igi homozygous Aicda-Cre⁺ Bcl6flx/flx mice do not display any GC B cells in any of the tissues examined, whereas their Cre-counterparts display robust GCs across all tissues (Fig. S1B). Both Cre⁺ and Cre- 564Igi homozygous groups harbor nearly exclusively idiotype positive B cells, whereas C57Bl/6J controls are negative (Fig. S1C). A low level of splenic plasmablasts and plasma cells are seen across all groups (Fig. S1D).

Of note, it can be hard to compare directly between bone marrow chimeras and non-chimeras. After the irradiation resets everything, 564Igi-initiated GCs emerge between 4 and 6 weeks (as shown in Figure 1), and plasmablasts and plasma cells may lag behind. Nonetheless, the expanded dataset in Supplementary Fig. 1 demonstrates the absence of spontaneous GCs in WT C57Bl/6J mice, 2) confirms the efficiency of Aicda-Cre Bcl6flx/flx genetic blockade of GCs, and 3) demonstrates the fidelity of the GC B cell and anti-idiotype stainings.

As additional negative controls, we have shown on multiple occasions, including in this study, that in mixed chimeras in which the 564Igi initiator bone marrow lacks the kappa chain (e.g., Figure 2 of (Degn et al., 2017), and this study Figure 1), or where no 564Igi initiator bone marrow is present (e.g., this study, Figure 3L-R, No Ab control, i.e., untreated non-564Igi chimeras), only very low baseline germinal center and plasma cell responses are observed.

We still think the most appropriate comparison is between the Cre⁺ and Cre- bone marrow chimera groups (between chimeras), and we find the representation in main Figure 2 simpler and more easily accessible than that in new Supplementary Fig. 1. We do acknowledge the value of the information presented by the added controls and have therefore created a new supplementary figure (new Supplementary Figure 1) to include these, rather than expanding the main figure, which is already at the size of a full page.

Specifically for Figure 3, we compare FoxP3-Cre Bcl6flx/flx mice, which were either not injected, injected with mIgG, Idiotype antibody, or anti-idiotype antibody (A-K), and wild-type chimeras, which were either not injected or injected with Idiotype antibody (L-R). However, here we understand that by **positive controls**, the Reviewer refers to controls that do present with germinal centers.

Of note, we always have an internal positive control for the GC population by virtue of the mesenteric lymph nodes, which invariably have a robust GC population directed against the commensal microbiota of the gut (see MesLN in panels E and P of Figure 3). This verifies the fidelity of our flow cytometry staining in terms of identifying this population when present.

We unfortunately do not know of a 'control antibody' that we could inject in naïve mice to induce robust germinal centers. However, as biological controls for Figure 3A-K, we do have data on the 564Igi line crossed to FoxP3-Cre Bcl6flx/flx background versus regular 564Igi heterozygous mice, which demonstrates robust GC B cell and idiotype B cell frequencies, and an exacerbated plasma cell output upon Tfr block (note that these are heterozygous 564Igi, whereas the Aicda-Cre Bcl6flx/flx biological controls above were homozygous 564Igi, hence the differences in idiotype positive B cell frequencies, but both heterozygous and homozygous 564Igi are robust positive controls). These data have now been included in new Supplementary Figure 1 (panels I-M).

This expanded dataset demonstrates that, 1) robust GC and plasmablast/cell frequencies are observed in Tfr blocked mice in the presence of an autoreactive knock-in BCR, and 2) demonstrates the fidelity of the GC B cell and anti-idiotype stainings.

While acknowledging the value of the added controls and their contribution to verifying our findings, we think it would confuse more than add clarity to include these groups on the presented graphs in the main figure, since Figure 3 is also already very expansive, so we prefer to include this as supplementary information (new Supplementary Figure 1).

Finally, as biological controls for Figure 3L-R, we consider all the BM chimera experiments where the BM inoculum includes 564Igi bone marrow, which invariably results in robust GC, plasma cell responses and autoantibodies (e.g., Figures 1, 2, 4, and 5 of this manuscript, as well as (Akama-Garren et al., 2021; Degn et al., 2017; Firl et al., 2018; van der Poel et al., 2019)).

Further, throughout the study germinal centres, and sometimes plasma cells were used as a surrogate marker for autoimmunity. I would prefer to see evidence that GCs are surrogates for autoimmunity in the different experiments and conditions. Would it be possible to show this, e.g. by presence of autoreactive antibody?

We have previously shown that the model presents with robust autoantibodies and epitope spreading, and we have directly demonstrated autoreactive wild-type derived B cell clones in a large fraction (3/4) of GCs interrogated (Degn et al., 2017). This is likely an underestimate, given that a low number of clones were screened per GC and the fact that many GC B cells have immeasurable antigen affinity even for defined model antigens with immunodominant epitopes. We have shown that the wild-type derived GC B cells are TLR7 addicted, in line with their autoreactive nature (Degn et al., 2017; Green et al., 2021). Subsequent work has shown clonal expansion and selection effects in the Tfh repertoire (Akama-Garren et al., 2021). In the present manuscript, we have shown anti-dsDNA autoantibodies in experiments presented in Figures 2, 3, 4, and 5. We have shown epitope spreading in Figure 6.

To our knowledge, with existing methodology it is impossible to definitively determine if a given autoantibody arose in a germinal center or an extrafollicular focus. Our previous observation that at least 3/4 GCs contained autoreactive clones in the model substantiate that the observed GCs are an expression of the autoreactive environment. This is corroborated by the aforementioned observations that in mixed chimeras where the 564Igi

initiator bone marrow lacks the kappa chain (e.g., Figure 2 of (Degn et al., 2017), and this study Figure 1), or where no 564Igi initiator bone marrow is present (e.g., this study, Figure 3L-R, No Ab control, i.e., untreated non-564Igi chimeras), only very low baseline germinal center and plasma cell responses are observed.

For this revision, we have additionally expanded the epitope spreading analyses presented in Figure 6 to also include serum antibody pull-downs of spleen extracts followed by MS analysis. These data have been included as new Figure 7 and new Supplementary Fig. 8. Of note, the feasibility of this approach demonstrates that (at least a subset of) the autoantibodies have a considerable affinity, indicating that they likely originate from GC responses, which display a higher degree of affinity maturation.

Finally, whereas the initiating 564Igi clone favors DNA and discrete nucleic acid associated targets, both in the original epitope spreading analyses (Degn et al., 2017) and in the ones presented in Figure 6 and the new added MS based analyses in new Figure 7 of the present manuscript, we are seeing expanded targeting of numerous protein antigens more typical of a GC response in our model.

Nonetheless, in recognition of the Reviewer's valid point and to avoid overstating our findings, we have made clear in the results section that we cannot definitively determine if a given autoantibody originated in a germinal center or an extrafollicular focus.

Minor issues:

Some figures are very small and difficult to read when printed.

Our apologies for this. It is a very data rich manuscript, and several of the figures are very busy, limiting the size of individual panels. If accepted, we will of course work with the copy-editing service to ensure that all figures comply with figure guidelines in terms of sizing and resolution. Of note, by rotating panels in Supplementary Figure 2, we already managed to increase the size of imaging panels considerably, as also outlined in our response to Reviewer 1.

The graphical abstract is not clear for someone who is not expert or has not read the manuscript. Can a legend be added? E.g. what are the pink star-shaped cells?

We realized that *Nature Communications* actually does not allow a graphical abstract. We propose to include this instead as summary Figure 8. We have now included a graphical legend and a brief legend explaining the figure to make it more immediately accessible.

Fig. 1A legend: numbers under spleens: unit is missing

We apologize for this error. We have now specified that numbers represent 'weeks' (post reconstitution).

Line 147: "+" is usually used for wt. Would another symbol for knockin be better?

This is a good point. We have changed the nomenclature to ki/wt instead of +/- for clarity. So a 564Igi homozygous mouse is termed 564Igi H^{ki/ki} K^{ki/ki}, etc. We have modified the explanation of the nomenclature on page 6 and corrected this throughout the manuscript and in Figure 1.

Line 148: K-/- means Ig kappa knockout?

No, this actually was meant to signify absence of the knock-in kappa chain, i.e., wild-type. However, we realize now that this could be great cause for confusion, so as per our response to the preceding comment, we have followed the Reviewer's suggestion to change the nomenclature and instead now call this 564Igi K^{wt/wt}. We thank the Reviewer for the suggestion to improve the nomenclature.

Line 148 and Fig. 1B: I assume transfer was K+/+ "or" K-/- B cells. Line 148 states "and".

Yes, "or" is clearer. Thank you for the suggestion to improve clarity. We have changed "and" to "or" in the sentence in question.

Fig. 2A: Bcl: "flx" is missing

We had omitted flx/flx for brevity but concede that it should be included for completeness and clarity. This has now been corrected in Figure 2A, as well as in Figure 3A where we had a similar issue (for FoxP3-Cre Bcl6flx/flx).

Suppl. Fig. 1G-I: I presume the GCs do not contain CD45.2 564Igi cells. Can this be shown?

We have previously shown in the mixed chimera model that 6+ weeks following reconstitution of mixed chimeras, 564Igi B cells contribute very little to the GC B cell population and WT B cells have largely taken over (Figure 2H of (Degn et al., 2017)). In the cohort presented in Suppl. Fig. 1 (now 2) in the present manuscript, we cannot definitively distinguish the 564Igi compartment from the MHCII^{null/null} or MHCII^{b/null} compartments as both are only CD45.2. We could stain for the Id receptor, however, it would not be possible to exclude that some 564Igi receptor positive cells had hypermutated away from Id reactivity. Furthermore, we cannot exclude that in an environment where a larger fraction of cells is unable to enter, there would be an increased representation of 564Igi cells. The low, but appreciable fraction of CD45.2+ GC B cells in Supplementary Figure 2 (previously 1) could hence be explained as, 1) a consequence of 564Igi B cell entry, 2) a consequence of transient but abortive entry of MHCII null cells (similar to that proposed in (de Vinuesa et al., 2000)), or 3) a combination of the two. Such considerations notwithstanding, based on the robust difference observed between the MHCII^{null/null} and MHCII^{b/null} groups, we believe our conclusion still stands, that inclusion of the wild-type compartment requires MHCII. If anything, were we able to subtract potential CD45.2 cells deriving from the 564Igi compartment, the difference would be even more dramatic, which would only further strengthen our conclusion.

Suppl. Fig. 3B legend: The time 6-8 wk and 12 wk do not fit with the single interval given in Fig. 4A, S3A, which is 8 wk. Why is that?

We thank the Reviewer for drawing our attention to this and apologize for the lack of clarity and the errors in the legends of Supplementary Figure 3B (now 4B), which said 10-13 weeks instead of correctly 10-16 weeks for euthanasia; in Main Figure 4B/C legend, which said 6 weeks instead of correctly 6-8 weeks; and in Main Figure 4D, which said 10-16 weeks instead of correctly 7-16 weeks. The indicated time points in the main text were correct. To clarify: for immunophenotyping, this was done at the same time point (7 weeks) for most of the b/d and d/d chimeras, but some of the d/d chimeras were typed at 8 weeks and some of the b/d chimeras were typed at 6 weeks (so 6-8 weeks overall). Euthanasia, blood sampling for serology, and flow analyses were performed in the interval from 7-13 weeks for b/d chimeras and 10-13 weeks for d/d chimeras, with a subset analyzed at 16 weeks. This was because of the practical aspect of analyzing some of the d/d chimeras on the Rhapsody platform in addition to the flow cytometry analyses, as well as our interest to evaluate whether the chimerism was stable in the d/d chimeras over a longer timespan (which we confirmed). For simplicity, the schematic overviews of experimental setups invariably say “6+ weeks”, as we have found analyses to be comparable in mixed chimeras following this time-point (at least within the investigated timeframe).

Reviewer #3 (expert in antigen presentation by B cells):

The authors present compelling evidence to prove that B cells initiate the autoimmune response outside of germinal center and epitope spreading occurs with antigen presentation dependent on MHC haplotype. The experiments are carefully designed to methodologically answer the scientific questions. Overall, the manuscript has valuable information that will promote the understanding of autoimmune diseases and the development of new treatment strategies. The article has some errors and require clarifications before acceptance for publication, as listed below:

1. It will be helpful to add a color-coded key to the graphs in figures 1, 2 and 4. Although it is evident that the colors are chosen according to the representation in the experiment schematic, adding the key for each graph will make interpretation easier for the readers.

We appreciate the Reviewer’s point that this would be clearer. We tried to accommodate this in Figures 2 and 4, but due to space constraints it is not possible to fit the legends into all the panels in the figures. We think it would look strange to have panel legends in only some of the panels and in only some of the figures. As a compromise, we have instead added a clearer color-coded key in panels A of Figure 2 and 4, as well as in new Supplementary Figure 1, panels A and I.

2. In Figure 2, panel Q/R show the images from Cre- chimera, however the text in the result section states “.....verified the absence of CD45.2 B cells from GCs of Cre+ chimeras (Fig. 2O-R)”. This needs to be rectified. [Also, I don’t know if this was intentional, but there is no panel W in the figure].

We apologize for the lack of clarity here. Panel Q and R were correctly labeled as Cre-, and in referring to these we simply meant the illustration of quantification that was the basis of the

quantified data in O and P, supporting the conclusion that CD45.2 B cells are absent from GCs of Cre⁺ chimeras. We have now included an additional high-magnification micrograph of a GC from a corresponding Cre⁺ chimera as new panel S (as also requested by Reviewer 1) and updated the in-text reference for increased clarity. We apologize for leaving out “W” (it is neither in the Swedish nor the Danish alphabet native to the first and last author, respectively). Panel lettering has been corrected and panels have been reordered.

3. In Figure 3, panel R has data showing Id antibodies in serum, however the text in the result section and the legend states “anti-dsDNA levels were comparable between the groups (Fig. 3Q and R)” and “Total IgG anti-dsDNA antibodies in sera of mice presented in panels M-Q” respectively. This needs to be rectified. [Also in the figure legend the panel “R” is mislabeled as “M”]

We thank the Reviewer for drawing our attention to these unfortunate errors. We have revised the sentences in question in the results section and legend and corrected the panel label.

4. In the discussion section the authors state that “Furthermore, our findings suggest that narrow antigen-specific targeting of T cell autoreactivity will fail, and at the same time emphasize the importance of moving towards antigen-specific targeting of B cells to curb autoimmunity”. As this is one of the main take away messages from this manuscript the authors should discuss this in more detail i.e. by talking about current therapies that may or may not follow this approach and what could be the possible hurdles/shortcomings of this approach.

We thank the Reviewer for this suggestion. Since the text is already quite expansive, we have had to be relatively conservative and only added 4 lines of text in the discussion and a reference to a very recent and extensive review on this topic, which puts us just above the 70 references limit recommendation.

We would like to end by again thanking all three Reviewers for their careful and critical evaluation of our work, which we feel has significantly improved our manuscript.

References included in this point-by-point response:

- Akama-Garren, E. H., van den Broek, T., Simoni, L., Castrillon, C., van der Poel, C. E., & Carroll, M. C. (2021). Follicular T cells are clonally and transcriptionally distinct in B cell-driven mouse autoimmune disease. *Nat Commun*, *12*(1), 6687. <https://doi.org/10.1038/s41467-021-27035-8>
- Baumjohann, D., Preite, S., Reboldi, A., Ronchi, F., Ansel, K. M., Lanzavecchia, A., & Sallusto, F. (2013). Persistent antigen and germinal center B cells sustain T follicular helper cell responses and phenotype. *Immunity*, *38*(3), 596-605. <https://doi.org/10.1016/j.immuni.2012.11.020>

- de Vinuesa, C. G., Cook, M. C., Ball, J., Drew, M., Sunners, Y., Cascalho, M., Wabl, M., Klaus, G. G., & MacLennan, I. C. (2000). Germinal centers without T cells. *J Exp Med*, *191*(3), 485-494. <https://doi.org/10.1084/jem.191.3.485>
- Degn, S. E., van der Poel, C. E., Firl, D. J., Ayoglu, B., Al Qureshah, F. A., Bajic, G., Mesin, L., Reynaud, C. A., Weill, J. C., Utz, P. J., Victora, G. D., & Carroll, M. C. (2017). Clonal Evolution of Autoreactive Germinal Centers. *Cell*, *170*(5), 913-926 e919. <https://doi.org/10.1016/j.cell.2017.07.026>
- Firl, D. J., Degn, S. E., Padera, T., & Carroll, M. C. (2018). Capturing change in clonal composition amongst single mouse germinal centers. *Elife*, *7*. <https://doi.org/10.7554/eLife.33051>
- Green, K., Wittenborn, T. R., Fahlquist-Hagert, C., Terczynska-Dyla, E., van Campen, N., Jensen, L., Reinert, L., Hartmann, R., Paludan, S. R., & Degn, S. E. (2021). B Cell Intrinsic STING Signaling Is Not Required for Autoreactive Germinal Center Participation [Original Research]. *Front Immunol*, *12*(5184), 782558. <https://doi.org/10.3389/fimmu.2021.782558>
- van den Broek, T., Oleinika, K., Rahmayanti, S., Castrillon, C., van der Poel, C. E., & Carroll, M. C. (2023). Invasion of spontaneous germinal centers by naive B cells is rapid and persistent. *bioRxiv*. <https://doi.org/10.1101/2023.05.30.542805>
- van der Poel, C. E., Bajic, G., Macaulay, C. W., van den Broek, T., Ellson, C. D., Bouma, G., Victora, G. D., Degn, S. E., & Carroll, M. C. (2019). Follicular Dendritic Cells Modulate Germinal Center B Cell Diversity through FcγRIIB. *Cell Rep*, *29*(9), 2745-2755 e2744. <https://doi.org/10.1016/j.celrep.2019.10.086>
- Wardemann, H., Yurasov, S., Schaefer, A., Young, J. W., Meffre, E., & Nussenzweig, M. C. (2003). Predominant autoantibody production by early human B cell precursors. *Science*, *301*(5638), 1374-1377. <https://doi.org/10.1126/science.1086907>

REVIEWERS' COMMENTS

Reviewer #1 (Remarks to the Author):

The authors have adequately addressed the concerns raised in the previous reviews.

Reviewer #2:

[Editor: This reviewer was no longer available for review and therefore replaced by Reviewer #4].

Reviewer #3 (Remarks to the Author):

The authors have modified the manuscript to my satisfaction, and I believe their responses to the previously raised comments and suggestions have been addressed appropriately. These changes have improved the manuscript significantly.

Reviewer #4 (Remarks to the Author):

The authors have adequately addressed reviewer #2's comments to my satisfaction. The issues relating to negative controls for fig 2 and positive controls for fig 3 are noted and I appreciate the effort and good faith shown by the authors to address the concern. The additional data is welcome and the edits have improved the manuscript by not overstating.

I would suggest though that the paper is very data heavy and was a real struggle to read especially as some of the figures contain too many panels. For example, fig 2 has 25! I appreciate that this may be for some journal formats but perhaps breaking up the 7 figures into 10 if allowed by the editor will make this much more accessible to the average reader.

RESPONSE TO REVIEWERS' COMMENTS

We would like to thank the Reviewers for their positive reception of our revised manuscript.

Below we provide a point-by-point response to the few comments presented for the revised version.

REVIEWERS' COMMENTS

Reviewer #1 (Remarks to the Author):

The authors have adequately addressed the concerns raised in the previous reviews.

We thank the Reviewer for their time and expertise in evaluating the revision.

Reviewer #2:

[Editor: This reviewer was no longer available for review and therefore replaced by Reviewer #4].

Reviewer #3 (Remarks to the Author):

The authors have modified the manuscript to my satisfaction, and I believe their responses to the previously raised comments and suggestions have been addressed appropriately. These changes have improved the manuscript significantly.

We are glad to hear that the Reviewer appreciated our additional work and considers the revision satisfactory.

Reviewer #4 (Remarks to the Author):

The authors have adequately addressed reviewer #2's comments to my satisfaction. The issues relating to negative controls for fig 2 and positive controls for fig 3 are noted and I appreciate the effort and good faith shown by the authors to address the concern. The additional data is welcome and the edits have improved the manuscript by not overstating.

I would suggest though that the paper is very data heavy and was a real struggle to read especially as some of the figures contain too many panels. For example, fig 2 has 25! I appreciate that this may be for some journal formats but perhaps breaking up the 7 figures into 10 if allowed by the editor will make this much more accessible to the average reader.

We would like to thank the Reviewer for stepping in and evaluating our response to the comments of Reviewer 2.

We appreciate the suggestion to simplify the presentation by splitting some of the more data heavy figures. However, as it is now, the figures are structured so that a single figure

corresponds to a section with a central take-home message. Splitting some of the figures would disrupt this structure, and we feel this would not simplify the story line. Specifically for Figure 2, we have however, also based on feedback from the Editor, simplified the imaging part and in this process reduced the panel number from 25 to 23. Moreover, we have taken care to polish the figures in terms of font size and panel lettering to make them as easily accessible as possible. We hope that these small changes go some way to accommodate the Reviewer's point.

We would like to end by again thanking all the four Reviewers that were involved in the process for their careful and critical evaluation of our work, which we feel has significantly improved our manuscript.